# Graphdiyne oxide nanosheets display selective anti-leukemia efficacy against DNMT3A-mutant AML cells

Qiwei Wang [1,2,3,12], Ying Liu [4,5,6,12], Hui Wang[5,7,12], Penglei Jiang[1,2,3,12], Wenchang Qian[1,2,3,12], Min You[4,5,6], Yingli Han[1,2,3], Xin Zeng[1,2,3], Jinxin Li[1,2,3], Huan Lu[1,2,3], Lingli Jiang[1,2,3], Meng Zhu[1,2,3], Shilin Li[4,5,6], Kang Huang[5,7], Mingmin Tang[8,9], Xinlian Wang[4,5,6], Liang Yan [5,10], Zecheng Xiong[5,11], Xinghua Shi [5,7], Ge Bai[9], Huibiao Liu[11], Yuliang Li [11], Yuliang Zhao [4,5], Chunying Chen [4,5] ✉ & Pengxu Qian [1,2,3] ✉

DNA methyltransferase 3 A (*DNMT3A*) is the most frequently mutated gene in acute myeloid leukemia (AML). Although chemotherapy agents have improved outcomes for *DNMT3A*-mutant AML patients, there is still no targeted therapy highlighting the need for further study of how *DNMT3A* mutations affect AML phenotype. Here, we demonstrate that cell adhesion-related genes are predominantly enriched in *DNMT3A*-mutant AML cells and identify that graphdiyne oxide (GDYO) display an anti-leukemia effect specifically against these mutated cells. Mechanistically, GDYO directly interacts with integrin β2 (ITGB2) and c-type mannose receptor (MRC2), which facilitate the attachment and cellular uptake of GDYO. Furthermore, GDYO binds to actin and prevents actin polymerization, thus disrupting the actin cytoskeleton and eventually leading to cell apoptosis. Finally, we validate the in vivo safety and therapeutic potential of GDYO against *DNMT3A*-mutant AML cells. Collectively, these findings demonstrate that GDYO is an efficient and specific drug candidate against *DNMT3A*-mutant AML.

Clonal hematopoiesis (CH) is an age-associated expansion of hematopoietic stem cells (HSCs) with acquired genetic or epigenetic changes that correlate with an increased risk of hematological malignancies including AML and myelodysplastic syndrome[1,2]. Both CH and AML exhibit the presence of recurrent somatic mutations in epigenetic modifiers, most frequently within the *DNMT3A* gene[3,4]. Loss of *Dnmt3a* in murine HSCs resulted in DNA hypomethylation and upregulation of HSC multipotency genes, which clonally expands the pre-leukemic HSC pool and increases the risk of hematopoietic malignancies, including AML[5,6]. In clinical studies, *DNMT3A* mutations have been associated with shorter overall and recurrence-free survival in AML[7]. Thus, *DNMT3A* represents one of the most important tumor suppressor genes in hematological malignancies, highlighting the need to

improve our understanding of its basic biological function(s) and develop new strategies for targeting it, especially in the context of *DNMT3A* mutations.

Currently, adult AML patients with *DNMT3A* mutation are usually treated with a combination of daunorubicin, a DNA damaging anthracycline drug and cytarabine, a DNA damaging antimetabolic agent[8,9]. However, several studies have reported that *DNMT3A* mutations promote anthracycline resistance via impaired nucleosome remodeling[10], suggesting that alternative strategies, beyond DNA damaging chemotherapies, could be beneficial. In this context, small molecule inhibitors of DNA methyltransferase enzymatic activity, such as 5-azacytidine and decitabine, have been explored for the treatment of *DNMT3A*-mutant AML patients with inconclusive results[11]. Another

**Fig. 1 | Cell adhesion was predominantly enriched in *DNMT3A* mutant AML cells. a** Venn diagram of up-regulated BP terms in *DNMT3A*-mutant AML cells from both AML patients (TCGA: *DNMT3A*mut, *n* = 36; *DNMT3A*wt, *n* = 115) and AML cell lines (CCLE: *DNMT3A*mut, *n* = 4; *DNMT3A*wt, *n* = 30). **b** Survival curves of AML patients with high and low expression designated gene set, and *n* is the number of biologically independent samples. The survival analysis was performed by the log-rank (Mantel−Cox) test. **c** Cell adhesion assay for human AML cells seeded on fibronectin or collagen. *n* = 3 biologically independent experiments. **d** Cell-cell adhesion between human AML cells and HUVECs. **e** Representative flow cytometry results for cell-cell adhesion between human AML cells and HUVECs. **f** Representative fluorescent image for cell-cell adhesion between human AML cells and HUVECs. Scale bar, 50 μm. **g** Statistical results for cell-cell adhesion between human AML cells and HUVECs. *n* = 3 biologically independent experiments. The data were shown as the mean ± SD. Statistical significance was tested with One-way ANOVA. Source data are provided as a Source Data file.

therapeutic strategy is to inhibit targets and/or signaling pathways that are abnormally upregulated by *DNMT3A* mutations. For example, pharmacologic inhibition of DOT1 like histone lysine methyltransferase (DOT1L), an enzyme with increased activity in *DNMT3A*-mutant AML, efficiently eliminated mutant cells both in vitro and in vivo[12]. Despite decades of research and numerous efforts to develop targeted, small molecule-based therapies for AML, including *DNMT3A*-mutant AML, clinical results have been discouraging[13]. This suggests that modalities beyond small molecules may represent an important opportunity for developing targeted therapies for AML.

Here, to identify potentially targetable vulnerabilities in *DNMT3A*-mutant AML cells, we perform comprehensive bioinformatic and in vitro analyses, which reveal that cell adhesion-related pathways are predominantly enriched in *DNMT3A*-mutant cells. We screen multiple types of carbon nanomaterials and identify that graphdiyne (GDY) oxide (GDYO), a novel 2D carbon material, shows the strongest inhibitory effect on *DNMT3A*-driven AML leukemogenesis. Mechanistically, we discover that GDYO nanosheets interact with adhesion molecules, ITGB2 and MRC2, which facilitates cellular uptake of GDYO. Moreover, GDYO nanosheets bind to actin and disrupt the actin filaments organization, which eventually lead to differentiation and apoptosis in AML cells. Finally, we validate the therapeutic potential against *DNMT3A*-mutant AML cells and in vivo safety of GDYO.

## Results

### Increased cell adhesion is a key feature in DNMT3A-mutant AML cells

To explore potential therapeutic drugs against *DNMT3A*-mutant AML cells, we elucidated the difference between *DNMT3A*-mutant and wildtype (wt) AML cells at the transcriptome level. We used the RNA-Seq data from the The Cancer Genome Atlas (TCGA) and Cancer Cell Line Encyclopedia (CCLE)[14] databases, and found that cell adhesion-related pathway was predominantly enriched in both *DNMT3A*-mutatnt AML patients and cell lines (Fig. 1a), and inversely correlated with the survival of the patients (Fig. 1b). Three common cell adhesion-related genes were identified from both TCGA and CCLE databases and verified by Q-PCR that they were indeed highly expressed in human AML cell lines (Supplementary Fig. 1). We also identified three down-regulated pathways in *DNMT3A*-mutant AML cells, but none of them was significantly associated with the patients' prognosis (Supplementary Fig. 2). Furthermore, we analyzed RNA-seq datasets of the mouse LSK (Lin⁻Sca1⁺c-Kit⁺) cells isolated from mice with conditional knock-in of *Dnmt3a*R878H mutation[15], as a model of *DNMT3A*-mutant AML featuring a common R878H mutation, and mouse HSCs sorted from *Dnmt3a*-/- mice[5], as a model of complete loss of DNMT3A. We found that genes related to cell adhesion were up-regulated in both of those models (Supplementary Fig. 3). Together, we determined that expression of cell adhesion-related genes was increased in *DNMT3A*-mutant AML cells.

Cell adhesion molecules (CAMs) are located on the cell surface and responsible for cell-extracellular matrix (ECM) and cell-cell adhesion[16]. CAMs not only recognize specific ligands, such as fibronectin, collagen or laminin secreted in the bone marrow (BM)[17], but also activate signaling pathways related to cell proliferation, survival, and drug resistance, which subsequently lead to chemoresistance and recurrence of leukemia[18]. To confirm the high adhesion feature of *DNMT3A*-mutant AML cells, we pre-coated the cell culture dishes with fibronectin or collagen and incubated them with different types of AML cells. We measured the amount of adherent cells after removing of the non-adherent ones, and observed that *DNMT3A*-mutated AML cells (OCI-AML2 and OCI-AML3) adhered more strongly when compared to the five different types of *DNMT3A*-wt cells (Fig. 1c). In addition, we prelabelled the human umbilical vein endothelial cells (HUVECs) and AML cells with different fluorescent dyes, co-cultured them, and determined the number of AML cells attached to HUVECs by flow cytometry or fluorescence imaging (Fig. 1d). Consistently, *DNMT3A*-mutant AML cells were more adhesive to HUVECs than *DNMT3A*-wt cells (Fig. 1e–g and Supplementary Fig. 4). Taken together, these data suggest that cell adhesion is markedly upregulated in *DNMT3A*-mutant AML cells, resulting in a more adherent phenotype.

## GDYO nanosheets exhibit specific anti-leukemia efficacy against DNMT3A-mutant AML cells

Previous studies have reported that CAM inhibitors increase chemosensitivity of leukemia cells[19], however, most of them show high toxicity to normal cells including HSCs[20], and poor inhibitory activity. Recently, carbon nanomaterials have been shown to modulate cell adhesion and guide cell fate[21–23]. Therefore, to examine whether these materials affect viability of *DNMT3A*-mutant AML cells, we tested cytotoxicity of different carbon nanomaterials in OCI-AML2 and OCI-AML3 cell lines. The carbon nanomaterials we tested include multiple-walled carbon nanotubes (MWCNT), amino-modified MWCNT (MWCNT-NH₂), carboxyl-modified MWCNT (MWCNT-COOH), graphene (G), graphene oxide (GO), GDY and GDYO. Unexpectedly, we found that GDY and GDYO, significantly decreased cell viability compared with other carbon nanomaterials, whereas G, which has been reported to kill certain types of AML cells[22], was not active (Fig. 2a).

To examine structural features that give rise to observed differences between GO and GDYO, we performed detailed comparison between the two carbon nanomaterials. Transmission electron microscopy (TEM) images revealed that both GDYO and GO displayed a nanosheet-morphology with occasional folding (Supplementary

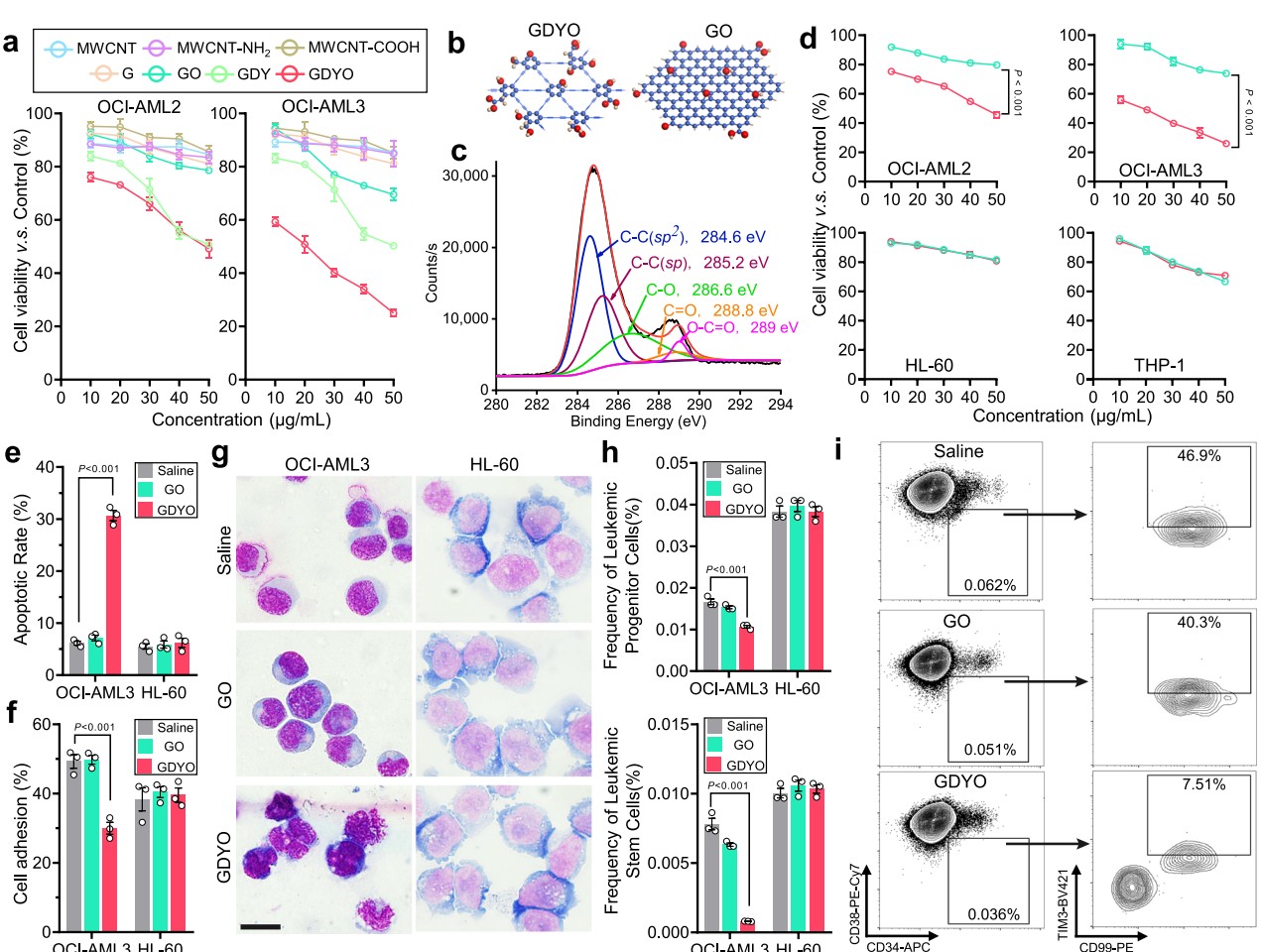

**Fig. 2 | GDYO showed anti-leukemia efficacy against *DNMT3A* mutant AML cells. a** Cell viability assay of OCI-AML2 and OCI-AML3 treated with different carbon-based nanomaterials for 24 h. *n* = 3 biologically independent experiments. **b** Schematic illustration showing the structures of GDYO and GO. Carbon and oxygen atoms were shown in blue and red respectively. **c** C 1 s spectra of XPS scan for GDYO. **d** Cell viability assay of *DNMT3A*-mutant AML cell lines (OCI-AML2, OCI-AML3) and DNMT3A wildtype AML cell lines (HL-60, THP-1) treated with GO or GDYO at different concentrations for 24 h. *n* = 3 biologically independent experiments. **e** Apoptotic rates in OCI-AML3 and HL-60 treated with 20 µg/mL GO/GDYO

for 48 h. *n* = 3 biologically independent experiments. **f** Cell adhesion assay on fibronectin for OCI-AML3 and HL-60 treated with 20 µg/mL GO/GDYO for 24 h. *n* = 3 biologically independent experiments. **g** Wright-Giemsa staining showing signs of maturation in GDYO-treated OCI-AML3. Scale bar, 20 µm. **h–i** Flow cytometry analysis of leukemic progenitor cells and leukemic stem cells in OCI-AML3 and HL-60 treated with 20 µg/mL GO/GDYO for 72 h. *n* = 3 biologically independent experiments. The data were shown as the mean ± SD. Statistical significance was tested with a two-tailed, unpaired Student's *t* test. Source data are provided as a Source Data file.

Fig. 5a). The size distribution of GDYO and GO were measured by dynamic light scattering (DLS) showing that the average size of GDYO (255 nm) was larger than GO (164.2 nm) (Supplementary Fig. 5b). Atomic force microscope (AFM) images confirmed that GDYO had multilayer nanosheet structure with a size of 200-300 nm and a thickness of 2.9-7.3 nm, while the average thickness of GO was about 2.05 nm (Supplementary Fig. 5c, d). In contrast to GO, benzene rings of GDYO are connected by butadiyne linkages (Fig. 2b). We used X-ray photoelectron spectroscopy (XPS) to examine the structure of GDYO and observed that GDYO spectrum showed features consistent with the presence of four different types of carbon atom s orbitals C-C ($sp^2$), C≡C ($sp$), C-O, and C=O (Fig. 2c). Therefore, as expected, although GDYO and GO displayed morphological similarity, their chemical structures were distinct.

To compare their biological effects on leukemogenesis, we tested the cytotoxicity of GDYO and GO in multiple leukemia cell lines and found that GDYO potently suppressed the proliferation of *DNMT3A*-mutant AML cell lines, while its effect on *DNMT3A*-wt AML cell lines (HL-60 and THP-1) and acute lymphoid leukemia cell lines (Jurkat and Nalm6) was more modest (Fig. 2d and Supplementary Fig. 6). Furthermore, GDYO decreased cell adhesion ability, increased apoptosis, induced cell differentiation (as measured by CD11b/CD14 expression and morphology), and reduced number of colony formation unit (CFU) in *DNMT3A*-mutant but not in *DNMT3A*-wt cells (Fig. 2e–g and Supplementary Fig. 7, 8, 9a, b). Leukemia stem cells (LSCs) have been reported to participate in drug resistance and recurrence of AML[24,25], and GDYO significantly decreased the frequency of leukemia progenitor and stem cells in *DNMT3A*-mutant cells (Fig. 2h, i and Supplementary Fig. 9c). In contrast, GO exhibited little or mild impact on all the leukemia cell lines (Fig. 2d–I and Supplementary Fig. 7, 8). To further evaluate whether in vitro GDYO treatment could affect in vivo leukemogenesis, we transplanted GDYO-treated OCI-AML3 cells into sublethally irradiated NOD/SCID/IL-2Rγ-null (NSG) mice, and found that pre-treatment of GDYO markedly reduced bioluminescence signals of the luciferase-expressing OCI-AML cells and prolonged the survival of engrafted mice (Supplementary Fig. 10). In peripheral blood (PB) of mice injected with GDYO-treated OCI-AML3 cells, we detected decreased numbers of white blood cells (WBC) including monocytes and neutrophils as well as increased number of red blood cells (RBCs) and platelets (Supplementary Fig. 11). Finally, pre-treatment with GDYO profoundly mitigated both the frequencies and absolute numbers of leukemia blasts and leukemia stem cells, while elevating CD11b⁺/CD14⁺ matured cells in both BM and spleen (Supplementary Fig. 12, 13). Collectively, these results indicate that pre-treatment with GDYO specifically attenuated leukemogenesis of *DNMT3A*-mutant AML cells both in vitro and in vivo.

## GDYO nanosheets display better dispersion potential than GO in culture media

To understand the determinants of different biological activities, we next focused on characterizing some key physicochemical properties of GO and GDYO, such as hydrophilicity and solubility under physiological pH and ionic strength conditions. The HOMO (highest occupied molecular orbital) and LUMO (lowest unoccupied molecular orbital) of G/GO and GDY/GDYO were calculated initially (Fig. 3a). Compared to G/GO that is composed of the $sp^2$-C6 hexagons, GDY and GDYO bear the butadiyne linkages in hexagonal directions of benzene rings, and thus feature many triangular cavities and the $sp$-$sp^2$ conjugation (Fig. 3b). Acetylenic bonds possessed stronger electron-withdrawing property and attract electrons in oxidation groups, such as carboxyl, hydroxyl and epoxy[26]. This suggests that GDYO should exhibit higher electrophilicity, ionizability and solubility in water. To investigate these properties, we then determined the ζ-potential values of aqueous dispersions of GDYO and GO. The ζ-potential values of both GDYO and GO were sensitive to pH, but GDYO showed less ζ-potential

value than that of GO at the same pH (Fig. 3c), which might be attributed to the dissociation of hydrogen ions and formation of ionized oxidation groups[27]. Furthermore, we performed pH titration of aqueous dispersions of GDYO and GO (Supplementary Fig. 14a) to determine the concentration of ionized groups and found that GDYO had more ionized groups than GO at the same pH (Fig. 3d), lower p$K_a$ values than GO (Supplementary Fig. 14b). Collectively, these results suggest that GDYO nanosheets were more likely to ionize and consequently form more stable interactions with water molecules, in agreement with their higher affinity for $H_2O$[28]. Moreover, higher level of ionization of GDYO sheets could provide additional electrostatic repulsion, and prevent cation-induced complexation in the culture media[29]. To probe this further, we examined the aggregation of GDYO and GO in different dispersion solutions and found that the hydrodynamic radius of GO increased over the course of 120 minutes in both saline and cell culture media, whereas GDYO hydrodynamic radius remained constant (Fig. 3e). This agrees with the previous study which reported that GO was kinetically aggregated in cell media within 24 h[30]. Owing to better hydrophilicity and stronger inter-sheet electrostatic repulsive force, GDYO displayed better dispersion ability in cell culture media and was more likely to interact with the suspension-cultured leukemia cells, while GO nanosheets were aggregated and precipitated, thus losing the interactions with cells (Fig. 3f). Finally, TEM images validated that GDYO nanosheets were internalized by the cells and distributed in the cytoplasm, whereas most GO nanosheets were not (Supplementary Fig. 15). Overall, these data suggest that GDYO is more biologically active because of its unique physicochemical properties, especially better hydrophilicity and stronger inter-sheet electrostatic repulsive force, which drive better dispersion potential in culture media and improved cellular uptake. However, given that GDYO was selectively active against *DNMT3A*-mutant AML cells, and not universally active against all cell lines tested, we hypothesized that there are other, cell line-specific factors that impact GDYO bioactivity.

## GDYO nanosheets specifically and directly bind to ITGB2 and MRC2

To systematically examine whether GDYO interacts directly with specific CAMs, we developed a nanoparticle pull-down protocol built on liquid chromatography-mass spectrometry (LC-MS/MS) (Supplementary Fig. 16), and screened for GDYO binders. We identified 138 GDYO-interacting proteins (Fig. 4a and Supplementary Data 1). Of note, KEGG analysis revealed that these proteins were related to signaling pathways involved in cell adhesion and phagocytosis (Fig. 4b and Supplementary Fig. 17), consistent with previous studies that large-sized nanomaterial was frequently internalized by phagocytosis[31]. Additionally, gene set enrichment analysis (GSEA) suggested that phagosome pathway was enriched in AML LSCs compared to normal HSCs or blast cells (Supplementary Fig. 18), which might explain our findings that GDYO treatment significantly eliminated LSCs (Fig. 2h–i).

To further narrow down the list of possible GDYO-binding proteins, we examined the expression levels of candidate genes by reanalyzing RNA-seq data of *DNMT3A*-mutant and *DNMT3A*-wt AML cells from CCLE database[14]. We noted that ITGB2, the first phagocytic integrin to be characterized[32], and MRC2, a collagen receptor that serves to direct large collagen fragments to lysosomal degradation through phagocytosis[33], exhibited the highest expression levels and were located on cell surface in *DNMT3A*-mutant AML cell lines (Fig. 4c, d). We also analyzed the bisulfite sequencing data from AML TCGA database and found that promoter regions of both *ITGB2* and *MRC2* were hypomethylated (Supplementary Fig. 19a), which was associated with their high expression levels in *DNMT3A* mutant cells (Supplementary Fig. 19b). Furthermore, we found that expression of *ITGB2* and *MRC2* was much lower in normal HSCs compared with AML cells (Supplementary Fig. 20), and that normal human and murine hematopoietic cells barely expressed *MRC2* (Supplementary Fig. 21).

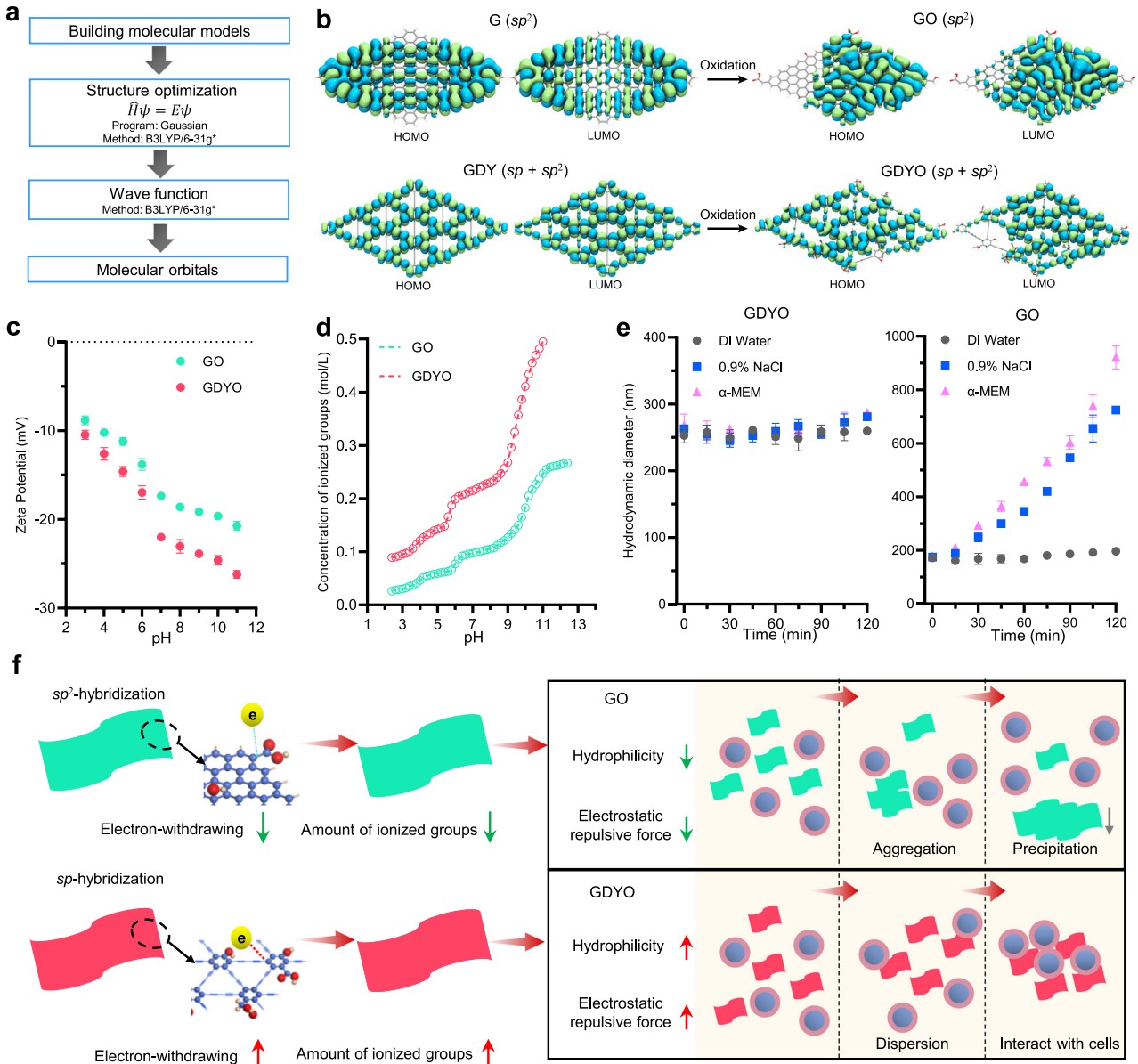

**Fig. 3 | GDYO displayed better dispersion potential than GO in culture media.** **a** Workflow of molecular orbital calculation. **b** Calculated HOMO and LUMO molecular orbitals of GO and GDYO. **c** Zeta potential of GO and GDYO nanosheets at different $p$H values. $n = 3$ independent experiments. **d** Concentration of ionized groups of GO and GDYO nanosheets at different $p$H values. **e** Hydrodynamic diameter of GO and GDYO nanosheets in different dispersion solutions. $n = 3$ independent experiments. **f** Schemes for dispersion kinetics of GO and GDYO in culture media. The data were shown as the mean ± SD. Source data are provided as a Source Data file.

This suggests that GDYO would have minimal toxicity in normal (healthy) cells.

To delineate the molecular mechanisms of the interaction at the nanosheet–protein interface, we performed molecular dynamic simulations to dissect the interactions between ITGB2 or MRC2 and monolayer GDYO nanosheet (Fig. 4e, g). By simulating the initial configurations and stable absorption configurations after 100 ns, we observed that GDYO nanosheet directly bound to specific domains within ITGB2 and MRC2 featuring, abundant hydrophilic and positively charged amino acid residues including lysine, arginine and histidine (Fig. 4f, h). The contact numbers showed that the interaction is stable, and the interaction energy analysis showed that the Van der waals (L-J) interaction is dominant for the charge-neutral models (Supplementary Fig. 22). Taken together, these observations suggest that GDYO binds ITGB2 and MRC2 directly, and this interaction facilitates attachment and entrance of GDYO into *DNMT3A*-mutant AML cells.

These above GDYO binders was identified in basal medium without serum, yet the previous in vitro cytotoxicity experiments were performed in the presence of serum, which meant that the surface of the incubated GDYO nanosheets had been covered by the serum proteins. In addition, intravenously injected GDYO will also adsorb plasma proteins in the follow-up in vivo studies. The protein corona may compromise delivery and targeting to the nanomaterials[34]. In order to prove whether GDYO can still bind to ITGB2 and MRC2 after the surface adsorption of proteins, we pre-incubated GDYO with 20% fetal bovine serum (FBS) or 20% mouse plasma solution, and the prepared protein-coated GDYO[B] or GDYO[M] was subsequently interacted with the plasma membrane protein solution of OCI-AML3, respectively (Fig. 4i). Interestingly, we found that GDYO[B] had increased binding affinity to MRC2 but no change to ITGB2 relative to bare GDYO, while GDYO[M] had significantly increased binding affinity to both receptors (Fig. 4j). To further investigate the source of this difference, we eluted

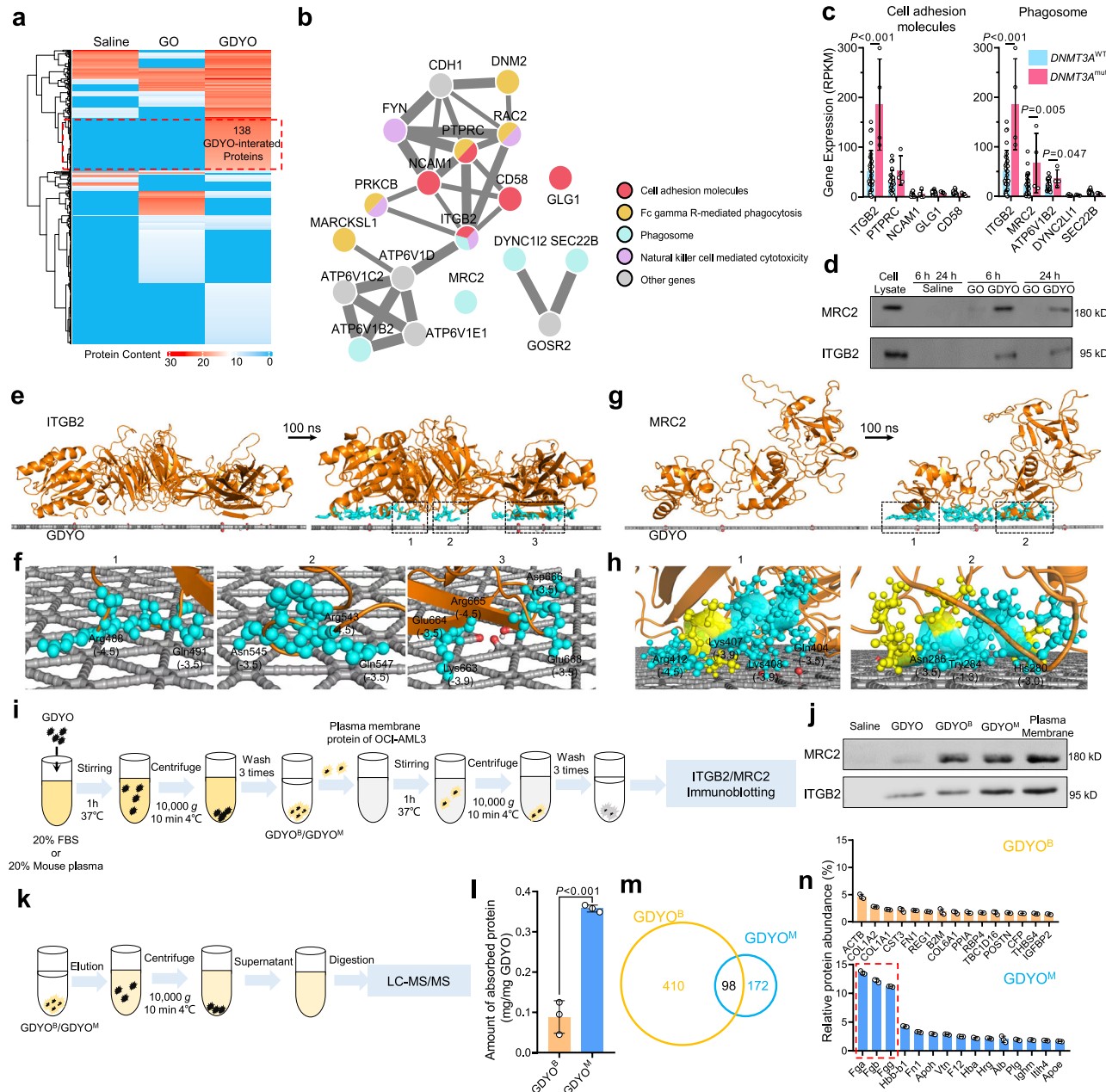

**Fig. 4 | GDYO directly interacted with ITGB2 and MRC2 in *DNMT3A* mutant cells. a** Heatmap of GDYO-interacted proteins detected by LC-MS/MS, *n* = 2. **b** Interaction network of key proteins. **c** Gene expression of proteins in key pathways between *DNMT3A*^mut and *DNMT3A*^wt human AML cell lines from CCLE database. *DNMT3A*^mut, *n* = 4; *DNMT3A*^wt, *n* = 30. **d** Antibody-guided immunoblot verified expression of selected proteins. **e**, **g** Side view of molecular dynamic (MD) simulation showing the adsorption of the ITGB2 (**e**) and MRC2 (**g**) proteins onto GDYO nanosheets. **f**, **h** The stable absorption residues of the ITGB2 (**f**) and MRC2 (**h**) proteins onto GDYO nanosheets. The numbers in bracket represent the hydrophobic parameters of corresponding residues. **i** Sample preparation diagram of to prepare GDYO^B/GDYO^M and to detect the interacted receptors. **j** Antibody-guided immunoblot for binding ability between ITGB2/MRC2 and GDYO/GDYO^B/GDYO^M. **k** Sample preparation diagram to profile proteins absorbed on GDYO^B/GDYO^M. **l** Protein content quantification of proteins absorbed on GDYO^B/GDYO^M. *n* = 3 biologically independent experiments. **m** Venn diagram of protein species of proteins absorbed on GDYO^B/GDYO^M. **n** Top 15 abundant proteins absorbed on GDYO^B/GDYO^M. *n* = 3 biologically independent experiments. The data were shown as the mean ± SD. Statistical significance was tested with a two-tailed, unpaired Student's *t* test. Source data are provided as a Source Data file.

proteins from GDYO^B and GDYO^M to perform protein profiling (Fig. 4k). We found that the quantity of total protein of GDYO^M was significantly larger than that of GDYO^B (Fig. 4l), while GDYO^B interacted with more proteins (Fig. 4m). This might be due to the fact that fibrinogen is present in plasma but not in serum, which together account for about 37% of the protein abundance in GDYO^M, including fibrinogen alpha chain (Fga), fibrinogen beta chain (Fgb) and fibrinogen gamma chain (Fgg) (Fig. 4n). Previous studies have reported that fibrinogen is one of ligands of ITGB2-containing integrins including

integrin α$_X$β$_2$ and α$_M$β$_2$, integrin α$_X$β$_2$ recognizes the sequence G-P-R in fibrinogen alpha-chain and integrin α$_M$β$_2$ recognizes P1 and P2 peptides of fibrinogen gamma chain[35,36], which could explain why GDYO^M exhibited higher binding affinity to ITGB2 than GDYO^B (Fig. 4j). Besides coagulation factors, the overlapped proteins in GDYO^B and GDYO^M were mainly extracellular matrix proteins, including collagens and fibronectin (Fig. 4n and Supplementary Fig. 23 and Supplementary Data 2). MRC2 is a receptor for collagen[33], hence both GDYO^B and GDYO^M had enhanced binding on MRC2 (Fig. 4j). To further investigate

whether the enhanced receptor interactions could affect anti-leukemia efficacy of GDYO against *DNMT3A*-mutant AML cells, we compared the anti-leukemia efficacies against OCI-AML3 of bare GDYO, GDYO^B and GDYO^M in serum-free media (Supplementary Fig. 24a). We found that GDYO^M exhibited more decreased cell viability, increased more apoptosis and cell differentiation, and showed more reduced number of LSCs and CFUs than bare GDYO in OCI-AML3, whereas GDYO^B did not (Supplementary Fig. 24b-i). In summary, the binding of GDYO to both ITGB2 and MRC2 is indispensable for its anti-leukemia effect. Fibrinogen and collagen in plasma will bind to the surface of GDYO nanosheets, increasing the interaction of GDYO on ITGB2 and MRC2, respectively, and thereby enhancing the anti-leukemia effect of GDYO against *DNMT3A*-mutant AML cells.

## GDYO nanosheets repress G-actin polymerization and disrupt actin cytoskeleton

*DNMT3A* is a DNA methyltransferase, and its mutations lead to global hypomethylation that promotes AML leukemogenesis[37]. To dissect downstream mechanisms underlying the inhibitory effects of GDYO, we first asked whether GDYO had an effect on DNA methylation by performing bisulfite sequencing in OCI-AML3 cells (Supplementary Fig. 25a, b). Surprisingly, the overall methylation was similar between PBS and GDYO group, implying that GDYO did not influence DNMT3A methyltransferase activity directly (Supplementary Fig. 25c-e). We then employed RNA-seq to assess the transcriptome change in GDYO-treated OCI-AML3 cells and identified 61 up-regulated genes and 508 down-regulated genes (Supplementary Fig. 26 and Supplementary Data 3). KEGG analysis identified "Regulation of actin cytoskeleton" as one of the most enriched down-regulated pathways in GDYO-treated cells (Fig. 5a and Supplementary Fig. 27, Supplementary Data 4). This suggest the possibility that actin cytoskeleton might be involved in the mechanism of GDYO cytotoxicity. Indeed, the mRNA levels of actin cytoskeleton associated genes were markedly reduced after GDYO treatment (Fig. 5b), including ROCK (Rho associated coiled-coil containing protein kinase), a serine/threonine kinase family that regulates the formation of actin stress fibers[38], and PI3K catalytic subunit p110α, which affects filamentous actin (F-actin) polymerization[39]. We further carried out western blotting analysis and confirmed that GDYO, and not GO, diminished the protein levels of ROCK1, ROCK2 and p110α in a dose-dependent manner (Supplementary Fig. 28).

To identify the direct target of GDYO in actin cytoskeleton pathway, we checked our protein LC-MS/MS data and noticed that actin might be the potential candidate (Supplementary Data 1). Indeed, pull-down experiments suggested that GDYO binds to actin in cell lysates of OCI-AML3 cells (Fig. 5c, d). We next performed molecular dynamics simulation between actin monomer and GDYO (Supplementary Fig. 29) and observed that GDYO bound to domains enriched for hydrophilic and positively charged amino acid residues, which were spatially adjacent to binding sites of latrunculin A (Fig. 5e, Supplementary Fig. 30), a compound that interacts with G-actin monomers to inhibit actin polymerization[40]. Furthermore, we performed in vitro actin polymerization/depolymerization assay, GDYO profoundly inhibited G-actin polymerization, with 1 μg of GDYO reducing the percentage of polymerized actin as low as latrunculin A (Fig. 5f). Increased amounts of GDYO (5 μg) depolymerized F-actin comparable to cytochalasin D (Fig. 5g), a compound that binds F-actin at barbed end to induce actin depolymerization[40], suggesting GDYO more readily interacted with actin monomers and prevented G-actin polymerization. Together, these data suggest that GDYO binds to actin directly, and prevents actin polymerization in *DNMT3A*-mutant AML cells.

To evaluate whether the cellular actin cytoskeleton was disrupted by GDYO, we transfected OCI-AML3 cells with Lifeact-tdTomato plasmid and performed live imaging of actin cytoskeleton. We observed that GDYO began to enter cells at 6 h, that large amounts of GDYO

nanosheets accumulated inside the cells at 24 h, and that F-actin contractile ring disassembled at 48 h (Fig. 5h). Furthermore, immunofluorescence staining data showed that GDYO partially localized into lysosomes at 12 h and the fraction of GDYO disappeared at 24 h (Supplementary Fig. 31). However, a significant amount of GDYO remained co-localized with actin at all time points (Fig. 5h), suggesting that GDYO interacted with actin after being released from the lysosome. Moreover, both immunofluorescence staining and live imaging of F-actin indicated that treatment by GDYO, reduced signals of cortical polymerized actin in OCI-AML3 cells, whereas GO had no effect (Supplementary Fig. 32, Supplementary Movie 1-3). This phenomenon was absent in *DNMT3A*-wt HL-60 cells (Supplementary Fig. 33, Supplementary Movie 4-6). Additionally, western blots showed that treatment of GDYO, reduced F-actin and increased G-actin level, and that GO did not affect actin (Fig. 5i). Perturbations of F-actin cytoskeleton have been reported to alter cell morphology and induce apoptosis[41,42]. Strikingly, scanning electron microscopy (SEM) revealed that cell surface became smooth and dented, and that CAMs were disappeared 24 h after GDYO treatment (Fig.5j), consistent with our previous findings that GDYO reduced cell adhesion and increased apoptosis in OCI-AML3 cells (Fig. 2e, f).

To determine if the inhibitory effects of GDYO on leukemogenesis were mediated by F-actin cytoskeleton disruption, we employed Jasplakinolide (Jas), an F-actin stabilizer, which could induce F-actin polymerization, to study its effect in GDYO-treated OCI-AML3 cells. Phalloidin staining showed that Jas effectively rescued effects of GDYO treatment, by restoring the cortical stabilization of F-actin cytoskeleton (Fig. 5k), rescuing cell apoptosis and differentiation (Fig. 5l and Supplementary Fig. 34), and increasing the number of CFUs (Fig. 5m). Collectively, these data provide strong support for a mechanistic model whereby GDYO binds to actin directly, thus preventing actin polymerization and disrupting F-actin cytoskeleton in *DNMT3A*-mutant AML cells.

## GDYO nanosheets possess in vivo therapeutic potential against DNMT3A-mutant AML cells

In light of the specific and effective inhibitory effects of GDYO, we explored whether GDYO could be harnessed as a potential therapeutic strategy against *DNMT3A*-mutant AML cells. To test this idea, we transplanted $1 \times 10^6$ OCI-AML cells into sublethally irradiated NSG mice via tail vein injection, and intravenously administered GDYO twice (Fig. 6a). Notably, GDYO treatment markedly eliminated leukemia cells at 14- and 21-day post-transplantation (Fig. 6b) and significantly prolonged the survival of engrafted mice (Fig. 6c). In PB, GDYO administration reduced numbers of leukocytes including monocytes and neutrophils, whereas elevated numbers of RBCs and platelets (Supplementary Fig. 35). In spleen, GDYO treatment reduced spleen weight and number of blast cells and leukemia stem cells, while increased frequencies of CD11b/CD14 mature cells (Fig. 6d, Supplementary Fig. 36, 37a-c). In BM, GDYO-treated mice exhibited normal morphology, and flow cytometry analyses showed that GDYO remarkably mitigated number of leukemic blast cells and LSCs as well as induced mature myeloid cell differentiation (Fig. 6f–i). In addition, neither obvious defects nor accumulation of GDYO was observed in other non-immune organs of engrafted mice (Supplementary Fig. 37d).

We next assessed the long-term toxicity by administration of GDYO into C57B/L6 mice, and observed no significant difference between PBS and GDYO-injected mice in body weight, number of BM long-term HSCs (LT-HSCs) (Supplementary Fig. 38), blood biochemical analysis and hematological analysis (Supplementary Fig. 39), as well as morphology in different organs (Supplementary Fig. 40a). Furthermore, in vitro hemolysis test showed GDYO nanosheets were biocompatible with whole blood (Supplementary Fig. 40b). GDYO treatment did not affect cell viability in mouse BM mononuclear cells (Supplementary Fig. 40c). Finally, long-term GDYO co-culture neither

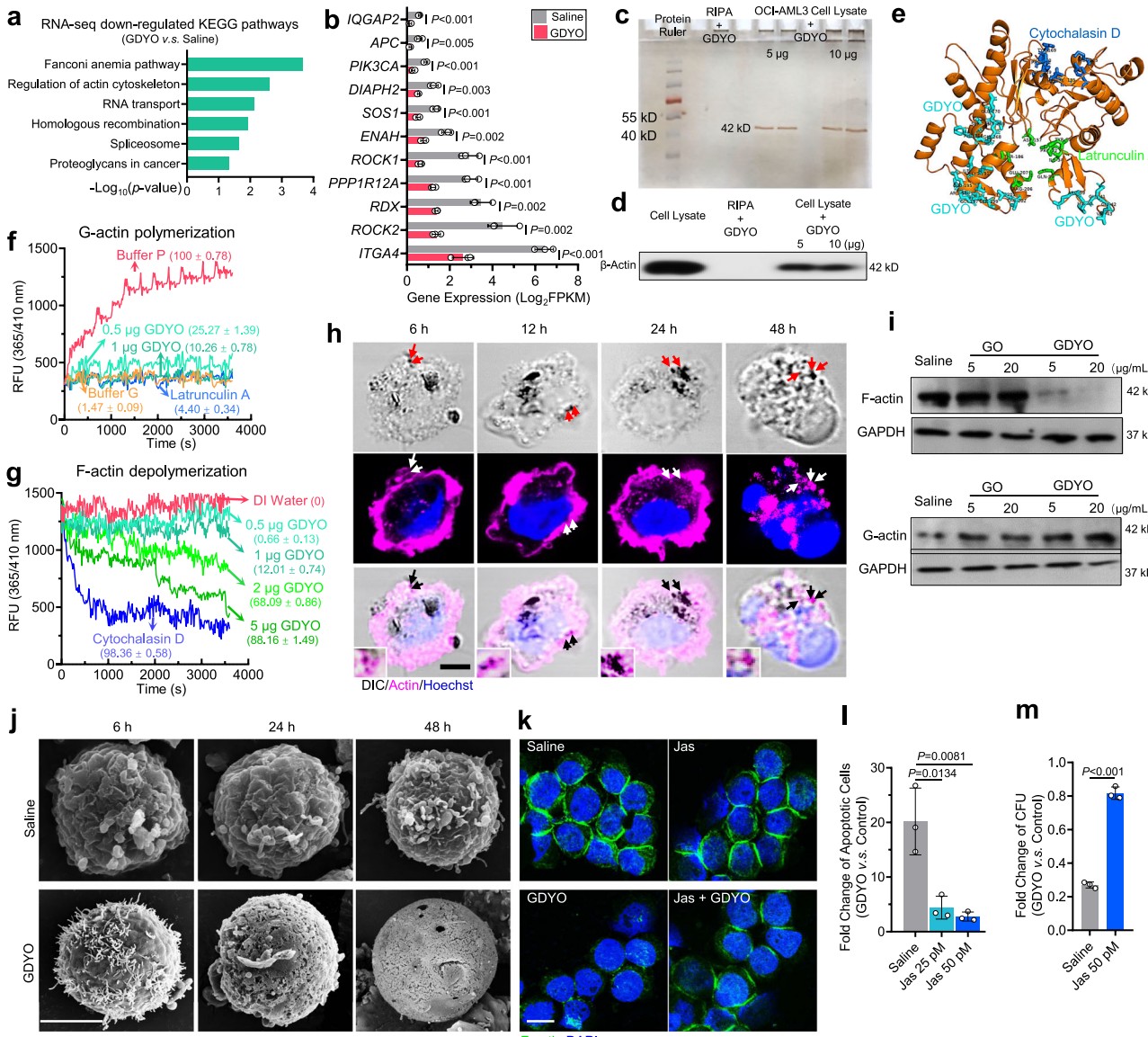

**Fig. 5 | GDYO dampened F-actin organization in *DNMT3A* mutant AML cells.**
**a** KEGG analysis of down-regulated genes in GDYO-treated OCI-AML3 cells.
**b** Fragments *Per* Kilobase *per* Million (FPKMs) of genes related in actin cytoskeleton in control and GDYO groups. *n* = 3 biologically independent experiments. **c** Silver staining of SDS-PAGE gel loaded with GDYO-treated cell lysates. **d** Antibody-guided immunoblot binding ability between β-actin and GDYO nanosheets. **e** Simulated binding site of GDYO (cyan), cytochalasin D (green) and latrunculin A (blue) onto the actin monomer. **f-g** G-actin polymerization (**f**) and F-actin depolymerization (**g**) curves treated with GDYO, cytochalasin D and latrunculin A. The numbers in brackets indicated the percentages of polymerization or depolymerization (%).

**h** Live cell imaging of actin with DIC component in GDYO-treated OCI-AML3 cells. Arrows indicated the co-localizations of GDYO nanosheets and actin. Scale bar, 5 μm. **i** Western blot analysis of F-actin and G-actin after GO or GDYO treatment. **j** Representative SEM images of OCI-AML3 treated by GO or GDYO for 6, 24 and 48 h. Scale bar, 5 μm. **k** Representative F-actin immunofluorescence confocal images of OCI-AML3 treated by saline, Jas, GDYO, Jas and GDYO for 48 h. Scale bar, 10 μm. **l-m** Fold change of apoptosis (**l**) and CFU (**m**) for GDYO *v.s.* control is rescued by Jas. *n* = 3 biologically independent experiments. The data were shown as the mean ± SD. Statistical significance was tested with a two-tailed, unpaired Student's *t* test. Source data are provided as a Source Data file.

decreased cell number nor induced apoptosis of LT-HSCs (Supplementary Fig. 40d-f). For other sensitive cells (*e.g.*, neurons), high concentration of GDYO also did not induce apoptosis (Supplementary Fig. 40g-h). In brief, these in vivo results suggest that GDYO possesses both promising therapeutic potential against *DNMT3A*-mutant AML cells and suitable biosafety profile.

## Discussion
In the present study, we found that cell adhesion-related genes were predominantly enriched in *DNMT3A*-mutant AML cells. By screening a series of carbon nanomaterials, we identified that GDYO showed strongest inhibitory effect on *DNMT3A*-driven AML leukemogenesis. Mechanistically, GDYO nanosheets displayed good dispersion

potential in culture media, and interacted with ITGB2 and MRC2, which facilitated cellular uptake of GDYO (Fig. 6j). Moreover, GDYO nanosheets repressed G-actin polymerization and disrupted actin cytoskeleton, leading to differentiation and apoptosis in AML cells (Fig. 6j). We also validated the in vivo safety and therapeutic potential of GDYO against *DNMT3A*-mutant AML cells, thus opening future opportunities for developing GDYO-based therapies. Importantly, GDYO may be effective in AML patients with *DNMT3A*[R882], who have an inferior outcome when treated with daunorubicin[43], likely due to the higher percentage of LSCs that exhibit properties of self-renewal and chemoresistance[24,44]. We observed that GDYO not only killed *DNMT3A*-mutant AML blast cells, but also eliminated the LSCs in both in vitro culture system and in vivo. Moreover, we found that GDYO had no

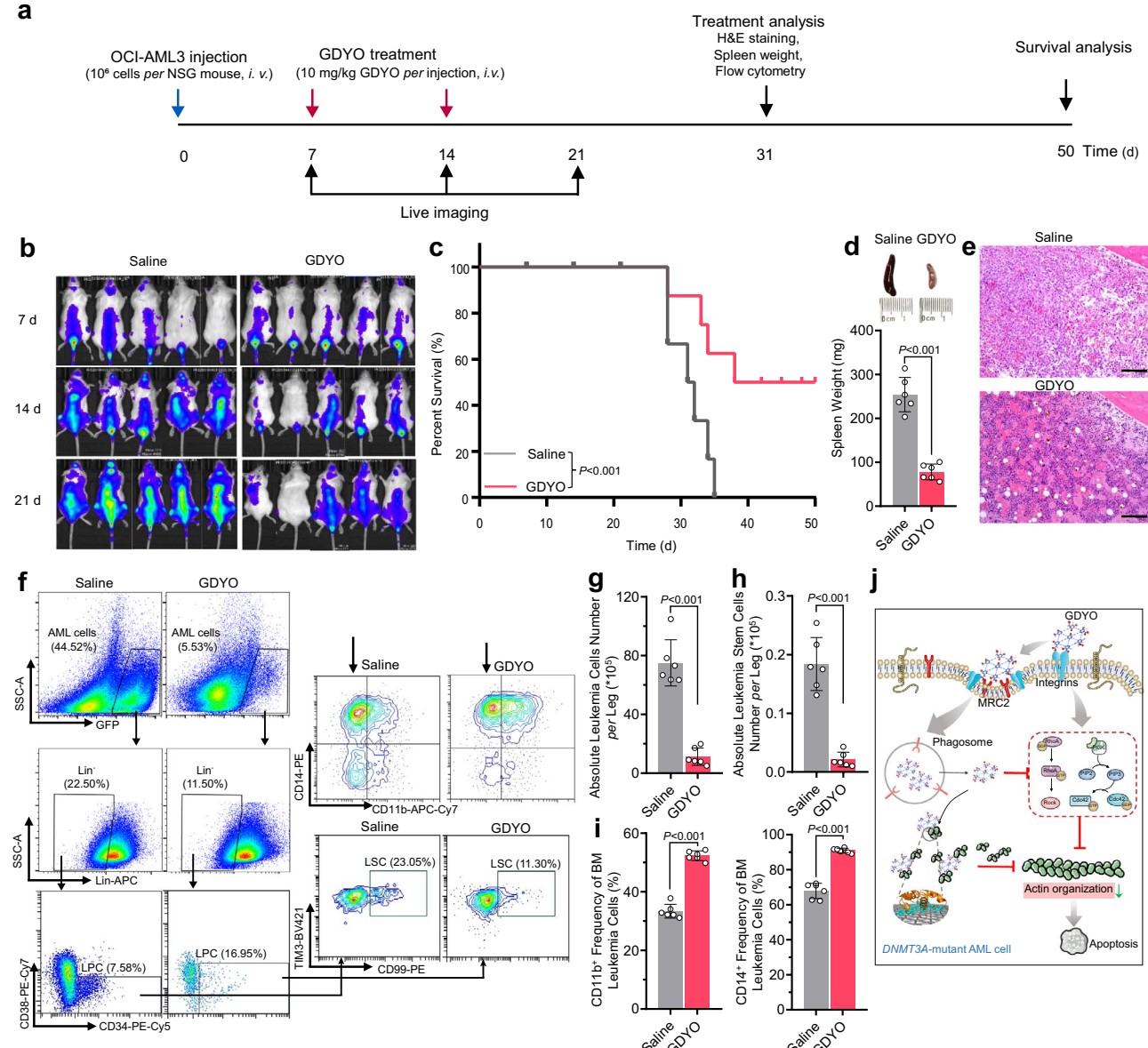

**Fig. 6 | GDYO exhibited in vivo therapeutic potential against *DNMT3A* mutant AML cells. a** Schematic illustration of the experimental design. **b** Representative IVIS spectrum images. *n* = 5 biologically independent animals. **c** Kaplan-Meier survival curve of AML mice injected with saline or GDYO. *n* = 8 biologically independent animals. **d** Spleen weight of AML mice injected with saline or GDYO at sacrifice. *n* = 6 biologically independent animals. **e** Bone marrow H&E sections of AML mice injected with saline or GDYO at sacrifice. Scale bar, 100 μm. **f** Representative flow cytometry images of BM total mononuclear cells. **g-i** Absolute number of leukemia cells (**g**) and absolute number of leukemia stem cells (**h**) in BM from AML mice injected with saline or GDYO. *n* = 6 biologically independent animals. **i** Myeloid differentiation (CD11b⁺/CD14⁺) of BM leukemia cells from AML mice injected with saline or GDYO. *n* = 6 biologically independent animals. **j** Illustration for GDYO therapy against *DNMT3A*-mutant AML cell. The data were shown as the mean ± SD. Statistical significance was tested with a two-tailed, unpaired Student's *t* test. Source data are provided as a Source Data file.

toxic effect on normal hematopoietic cells and other non-immune organs, suggesting the therapeutic potential of GDYO to treat *DNMT3A*-mutant AML patients in the future.

Our nanoparticle pull-down protocol and binding target analysis revealed that GDYO binds directly to proteins involved in signaling pathways that mediate organization of actin cytoskeleton (ITGB2 and MRC2, Fig. 4)[45,46], and reduced protein expression of ROCK and PI3K p110α (Supplementary Fig. 28). GDYO enhanced interaction on ITGB2 and MRC2 with mouse plasma protein absorption, and improved the anti-leukemia effect in vitro, which partially revealed the mechanism of effective therapeutic effect in mice AML model. Furthermore, GDYO binds directly to actin (Fig. 5e–g), resulting in major perturbations in the actin cytoskeleton (Fig. 5h–k). Destruction of actin cytoskeleton is

as one of the major triggers of apoptosis[47], and studies have also highlighted the pivotal roles of actin cytoskeleton in the initiation of apoptotic process[48,49]. Moreover, morphological alterations during cell differentiation are also modulated by actin cytoskeleton dynamics[50]. Consistently, our Jasplakinolide rescue data suggesting that disruption of actin cytoskeleton is the major downstream mechanism underlying the cytotoxic effects of GDYO on *DNMT3A*-mutant AML cells.

Previous studies have reported that GO, a 2D carbon material related to GDYO, binds to actin and induces perturbation to actin cytoskeleton in lung cancer cells[51,52]. However, our results showed that GO barely influenced either cell survival or actin cytoskeleton in AML cells. One possible explanation for this discrepancy is our observation

that GO nanosheets readily aggregate in cation-containing biological solutions, thus GO could neither contact suspended AML cells nor bind to core proteins associated with cell adhesion and phagocytosis. Moreover, recent studies suggested that GO was more cytotoxic to normal cells than GDYO due to the ion strength-related kinetic aggregation[30], and blood exposure to GO may cause anaphylactic death in non-human primates[53]. In stark contrast, GDYO selectively targeted *DNMT3A*-mutant AML cells but spared normal cells. Clinical applications will require further studies of GDYO pharmacokinetics, including the long-term persistence of GDYO in the blood and improvement of delivery methods[54]. Nevertheless, our work sheds important light on the anti-leukemia efficacy of GDYO against *DNMT3A*-mutant AML.

## Methods

### Cell culture

The human *DNMT3A*-mutated AML cell lines OCI-AML2 and OCI-AML3 were purchased from the Leibniz-Institute DSMZ (http://www.dsmz.de). The cell lines HL-60, Jurkat, Kasumi-1, THP-1, Molm-13, NB-4, K-562, Nalm6 and HEK-293T were obtained from the American Type Culture Collection (ATCC). Cells were cultured as recommended: OCI-AML3 and OCI-AML2 in α-MEM supplemented with 20% FBS; HL-60, K-562 and Nalm6 in IMDM supplemented with 10% FBS; Jurkat, Kasumi-1 and THP-1 in RPMI supplemented with 10% FBS. HEK-293T in DMEM supplemented with 10% FBS. Cell lines were tested and verified to be mycoplasma negative via PCR.

Mouse primary BM cells were harvested from femur and tibia of C57BL/6 J mice in PBS with 2% FBS. Red blood cells were lysed using a 0.16 M ammonium chloride solution, and the cells were filtered with 70 µm strainers to generate single cell suspensions. Harvested cells were cultured in StemSpan™ SFEM (Stemcell Tec.) supplemented with SCF (50 ng/mL) and TPO (50 ng/mL).

### Synthesis and characterizations of GO/GDYO

GDYO was prepared by acid-mediated oxidization of GDY by a modified Hummer's method[30]. In brief, 10 mg GDY power was added in the solution of concentrated 1 mL $HNO_3$ and 3 mL $H_2SO_4$ in a three-necked flask, and then 10 mg $KMnO_4$ was slowly added to the mixture under vigorous stirring in an ice bath. After cooling to room temperature, the mixture was transferred to an oil bath at 80 °C and vigorously stirred for 24 h to give a brown suspension. After being cooled to room temperature, the suspension was ultrasonicated for 10 min and the pH was adjusted to 8.0 with NaOH. Then, the product was collected by centrifugation at 10,000 $g$ for 10 min and washed with pure water for three times. The resulting product was re-dispersed into ultrapure water and ultrasonicated for about 12 h to obtain a homogeneous brown aqueous dispersion. The above solution was dialyzed (cutoff, 3500) for 7 days to give the pure GDYO for further use.

GO was prepared from expandable graphitic flake using a modified Hummer's method[30]. Typically, 0.5 g graphite and 0.5 g $NaNO_3$ were mixed together with 23 mL $H_2SO_4$ and the mixture was vigorously stirred in an ice bath, and then 3 g $KMnO_4$ was slowly added to the suspension. The above solution was transferred to a 35 °C water bath and vigorously stirred for 2 h to give a thick paste. Then, 40 mL water was slowly added to the reaction solution, and the mixture was stirred for 30 min, followed by raising the temperature to 95 °C. Next, 100 mL water was added the mixture and 3 mL $H_2O_2$ (30%) was slowly added, while the color of solution turned from dark brown to yellow. Finally, the suspension was filtered and washed with 1 M HCl and pure water for three times and dried in vacuum at 50 °C for 24 h to obtain GO.

GDYO/GO powder was dispersed in ultrapure water to 2 mg/mL with ultrasonication (500 W) at 30 °C for 1 h, and UV sterilization for further cell culture. Transmission electron microscopy (TEM) was performed using a FEI Tecnai G2 F20 U-TWIN TEM instrument. Scanning electron microscopy (SEM) was performed using a Hitachi

SU8220. Atomic force microscopy (AFM) was performed using a Bruker Multimode-8 AFM microscope in ScanAsyst mode. X-ray photoelectron spectra (XPS) was collected on a Thermo Fisher ESCALAB 250 XPS instrument. The hydrodynamic sizes and Zeta potentials of the nanoparticles were determined by dynamic light scattering (DLS) using a Malvern Zeta sizer Nano series Nano-ZS instrument.

### Quantitative PCR (Q-PCR)

Total RNA was isolated from different AML cell lines using Trizol (Invitrogen, US) according to the manufacturer's protocol, and quantified. After reverse transcribed into cDNA, a reaction system with a volume of 20 µL containing cDNA, primers and SYBR-green (Takara, Japan) was mixed. PCR was processed in ABI 7500 Q-PCR system as following procedure: Denaturation at 95 °C for 10 min, followed by 40 cycles of 95 °C for 15 s and 60 °C for 1 min. Gene-specific primers sequences are as follow:

ITGB2-F: TGCGTCCTCTCTCAGGAGTG
ITGB2-R: GGTCCATGATGTCGTCAGCC
RND3-F: CCCTCTCTTACCCTGATTCGG
RND3-R: TGGCGTCTGCCTGTGATTG
GPNMB-F: CCTCGTGGGCTCAAATATAACAT
GPNMB-R: ACTGTCCTCTGACCATGCTGT

All Q-PCR reactions were repeated three times, and the relative quantifications of gene expression were analyzed by the $2^{-\Delta\Delta Ct}$ method.

### Cell adhesion assay

For AML cell-ECM adhesion, fibronectin or collagen (corning) was pre-coated to 96-well plates overnight. $1 \times 10^4$ AML Cells were initially seeded and cultured for 30 min, then the non-adherent cells were washed by PBS. The adherent cells were further determined by CCK-8.

Cell adhesion (%) = Adherent cell ability/Unwashed cell ability × 100%

For AML cell-HUVEC adhesion, confluent CellTracker Red-stained HUVEC monolayers were first established on 24-well plates. Equal in number CellTracker Green-labelled AML cells were seeded and cultured overnight. After washing non-adherent cells, the rest of the cells were imaged by a fluorescent microscopy, digested by trypsin and quantified by flow cytometry.

Cell adhesion (%) = Adherent AML cell number/ Planted AML cell number × 100%.

### Cell viability assay

Cell viability was determined by CCK-8 assay. $1 \times 10^4$ cells were seeded overnight in 96-well plates. Cell viability was tested after 48 h incubation with GDYO or GO by measuring the absorbance at 450 nm.

### Flow cytometry analysis

Apoptosis was detected by Annexin V and PI staining. Cells were pre-treated with GDYO or GO (20 µg/mL) for 72 h and washed twice by pre-cooling PBS. $1 \times 10^5$ Cells were collected and resuspended by binding buffer, Annexin V and PI were added into cell suspension. After 20 min incubation, flow cytometry was detected in FITC and PE channel. For myeloid differentiation, cells were stained with antibodies against CD11b and CD14. For leukemia stem cells, cells were stained with antibodies against CD34, CD38, TIM3, CD99, together with human lineage cocktail (CD3, CD14, CD19, CD20, CD56). Antibody staining was performed at 4 °C for 45 min. Flow cytometry data was collected by Cytoflex LX, Beckman. Detail antibodies information were summarized in Supplementary Data 5.

### In vitro colony-forming unit (CFU) assay

Cells were pre-treated with GDYO or GO (20 µg/mL) for 12 h. 1,000 cells were seeded in methylcellulose media (H4034, Stemcell Technologies, Inc.) in triplicates. After 10 days, plates were scored for colony number on microscopy.

## Cytospin and Wright-Giemsa staining

Cells were collected, washed, and resuspended in PBS at a concentration of $10^5$/mL. A total number of $2 \times 10^4$ cells were loaded to perform cytospin (Thermo Scientific Shandon) at $150\,g$ for 1 min, and the cells were allowed to air dry. Cells were stained in 100% Wright-Giemsa (American MasterTech) for 5 min and washed in PBS buffer for 1 min. Stained cells were rinsed in deionized water, and coverslips were affixed with Permount prior to microscopy.

## pH-Titration

The concentration of ionized groups on GDYO and GO sheets at different pH values were determined using a pH titration as previously reported[28]. 0.1 g of GDYO/GO was taken in a beaker containing 20 mL of 0.1 M NaOH solution and 0.1 M HCl solution added in incremental steps of 0.25 mL. At each step the pH of the solution was recorded after ensuring that equilibrium had been attained. The experiment was repeated with the same volume of NaOH but now without additional GDYO/GO (Blank). The difference in the volumes of HCl in the titration curves for the same value of pH gave the concentration of the ionized groups *per* g of GDYO/GO at that pH.

## Mice

C57BL/6 J and NSG mice were purchased from Jackson Laboratory. Male and female mice from 8 to 12 weeks old were used for all studies. All mice were housed under specific pathogen-free conditions. All mice were cultured in suitable temperature and humidity environment and fed with sufficient water and food (25 °C, suitable humidity (typically 50%), dark/light cycle for 12 h). All animal experiments were performed according to protocols approved by the Institutional Animal Care and Use Committee (IACUC) of Zhejiang University School of Medicine (#ZJU20220027).

## In vivo transplantation studies

For pretreated experiment, OCI-AML3-GFP$^+$/Luciferase$^+$ was pretreated with GDYO or GO (20 μg/mL) for 12 h. $1 \times 10^6$ live cells were transplanted into NSG mice via tail vein injection. For GDYO treatment, $1 \times 10^6$ live OCI-AML3-GFP$^+$/Luciferase$^+$ were transplanted into NSG mice, and then GDYO was injected via tail vein at day 7 and 14 with a dose of 0.2 mg *per* mouse. Leukemia development was monitored by in vivo bioluminescence imaging with small-animal imaging system (IVIS Spectrum, PerkinElmer). The mice were *i.p.* injected with D-luciferin firefly (PerkinElmer) at a dose of 3 mg *per* mouse. The bioluminescence images were acquired 15 min after injection. Mice were randomly sacrificed at day 31 ($n = 6$), and then the spleen was removed and weighed. The femur and tibia were removed and analyzed by flow cytometry. The remaining mice ($n = 6$) were raised to calculate the survival curve up to 50 days.

## Hemavet Analysis

50 μL anticoagulated peripheral blood samples were collected from the submandibular vein of mice and then analyzed on the BC-2000 hematology analyzer (Mindray, China) programmed with mouse hematology settings.

## Biosafety analysis

In the animal test, GDYO suspended in PBS was administrated into C57B/L6 mice via tail vein injection (10 mg/kg body weight) every two days for 7 times. The body weight was monitored in 4 weeks and then the mice were sacrificed. The serum samples and main tissues (heart, liver, spleen, lung, and kidney) were gathered for the serum biochemical analysis and histopathological examination. The BM cells were isolated to detect the LT-HSC numbers. For LT-HSC surface phenotyping, a lineage cocktail (Lin) was used, including CD3, CD4, CD8, CD11b (Mac-1), Gr1, CD45R (B220), IgM and Ter119.

For the in vitro cell test, mouse BM cells were co-cultured with GDYO at different concentrations. Cell viability was tested after 48 h incubation with GDYO by measuring the absorbance at 450 nm. For long-term co-culture, mouse BM mononuclear cells were co-cultured with GDYO at 20 μg/mL, fresh media were replaced every three days. Number of LT-HSCs (Lin$^-$Sca-l$^+$c-Kit$^+$CD150$^+$CD48$^-$) was detected by flow cytometry. For neuron cytotoxicity assay, SH-S5Y5 cell line was treated with GDYO at 50 μg/mL, apoptosis levels were detected after 48 and 72 h. For hemolysis test, The GDYO were dispersed in 0.9% NaCl and mixed with 0.5 mL red blood cells for 8 h, with 0.9% NaCl as a negative control and red blood lysis (NH$_4$Cl 150 mM, NaHCO$_3$ 10 mM, EDTA 0.1 mM) as a positive control. After 8 hours of incubation, the mixture was centrifuged at $450\,g$ for 10 min. Detail antibodies information were summarized in Supplementary Data 5.

## Analysis of GDYO-interacted proteins

The experimental diagram to identify binding proteins in OCI-AML3 of bare GDYO was described in Supplementary Fig. 16. For details, cells were co-incubated with GO/GDYO for 24 h in basal medium without serum to avoid the protein pre-binding to GO/GDYO. Then cells were harvested and washed 3 times with PBS to clean up the free GO/GDYO. Cells were swelled and fractured with hypotonic solution (0.075 mol/L KCl, protease inhibitor cocktail). After centrifuge at $3,000\,g$, the supernatant containing cell-interacted GO/GDYO was transferred and centrifuged at $14,000\,g$. The pellet was washed 3 times with PBST and characterized with SDS-PAGE and silver staining. The experimental diagram to identify absorbed proteins in FBS or mouse plasma of GDYO was described in Fig.4i, k. For details, GDYO nanosheets were incubated with 20% FBS or 20% mouse plasma in saline with stirring for 1 h. After centrifuge at $10,000\,g$, the pellet was washed 3 times with PBST. Proteins in pellet were extracted with SDT lysis buffer (10% SDS, 100 mM Tris–HCl and 1 mM dithiothreitol, pH 7.6). After sonication for 15 min and boiling for 10 min, the solution was centrifuged at $14,000\,g$ for 30 min. Filter assisted sample preparation digestion was used for proteolytic digestion and high-performance liquid chromatography (HPLC) was employed for protein purification. Afterwards, LC-MS/MS was performed using a Q Exactive mass spectrometer (Thermo Fisher Scientific) coupled to an Easy nLC device (Thermo Fisher Scientific) for 60 min. Data analysis and protein identification were performed using MaxQuant v.1.3.0.5 software (Max Planck Institute of Biochemistry, Martinsried, Germany).

## Western blot

For GDYO-interacted ITGB2 and MRC2 detections, bare GO/GDYO or GDYO$^B$/GDYO$^M$ were incubated with cell lysate or plasma membrane protein with stirring, respectively. After centrifuge at $10,000\,g$, the pellet was washed 3 times with PBST. Proteins in the pellet extraction was performed by SDT buffer elution, sonicating and boiling, then the solution was centrifuged at $14,000\,g$ for 30 min and the supernatant was collected. For intracellular proteins extraction, OCI-AML3 were treated harvested and lysed in the RIPA protein extraction buffer; and for G-actin extraction, cells were lysed by soluble actin extraction solution (50 mM Tris-HCl, 300 mM Sucrose, 25 mM NaCl, 2 mM EDTA, 25 mM NaF, 1 mM NaCO$_3$, 0.2% Triton X-100 and protease inhibitor cocktail). The protein samples were mixed with loading buffer containing SDS, dithiothreitol, and bromophenol blue and boiled. Following separated by SDS-PAGE, proteins in the gel were transferred to a polyvinylidene difluoride (PVDF) membranes (Millipore, MA, USA). The membranes were blocked with 5% (w/v) nonfat powdered milk in TBST for 1 h at room temperature and incubated with the diluted primary antibody overnight at 4 °C. After washing three times with TBST, the membranes were incubated with the diluted horseradish peroxidase (HRP)-conjugated secondary antibody for 1 h at room temperature. After washing three times, the protein level was imaged by using

ImageQuant LAS 4000 system (GE, USA). Detail antibodies information were summarized in Supplementary Data 5.

## Lentiviral transfection and fluorescence-activated cell sorting (FACS)

Lentiviral vectors pWPXLd-Luciferase and pLenti-Lifeact-tdTomato were purchased from Addgene. HEK-293T cells (checked routinely for absence of mycoplasma contaminations) were kept in DMEM supplemented with 10% FBS. Lentiviral particles were prepared by transiently transfecting HEK293T with lentiviral vectors (10 mg/10 cm dishes) together with packaging vectors pMD2-VSVG (2.5 mg) and pPAX2 (7.5 mg) by using Lipofectamine™ 3000 (ThermoFisher) according to manufacturer instructions. After 6 h, transfection medium was changed. 48 h post-transfection, supernatant was collected, filtered through 0.45 μm cell strainer (Falcon) and stored at −20 °C. $1 \times 10^6$ OCI-AML3 cells were seeded in 3 cm dish containing 500 μL serum-free α-MEM and 500 μL viral supernatant. After 12 h, transfection medium was removed. 48 h post-transfection, cells were collected to sort fluorescence positive cells by FACS (moflo-Astrios EQ, Beckman). Sorted subpopulations were cultured in α-MEM supplemented with 20% FBS and then cryopreserved in liquid nitrogen.

## F-actin immunofluorescence staining and live imaging

Leukemia cells were attached to fibronectin-coated glass bottom dishes. After GDYO/GO treatment, cells were washed and fixed in 1% paraformaldehyde at room temperature for 15 min. Cells were then permeabilized with 0.2% Triton X-100 at room temperature for 30 min. To visualize F-actin, cells were stained for 30 min with Alexa Fluor 488 phalloidin (Invitrogen). For LAMP1 staining, cells were stained with primary LAMP1 antibody overnight and washed twice and stained with FITC-labeled secondary antibody. Cells were mounted on glass slides with Fluro-gel mounting media containing DAPI and covered with glass coverslips. Images were taken with a confocal microscope (Nikon A1R) using a 63×1.4NA oil objective. Detail antibodies information were summarized in Supplementary Data 5. Leukemia cell lines OCI-AML3 and HL-60 expressing Lifeact-tdTomato were attached to fibronectin-coated glass bottom dishes overnight. GDYO or GO were added at zero time point and TIRF images were taken over 48 h with an interval of 10 min. To quantify, multiple cells ($n = 20$ for each group) were photographed, and the fluorescence intensity of ROI was measured by ImageJ 1.48 v.

## TEM and SEM observation of AML cells

GDYO or GO-treated cells were soaked in the modified Karnovsky fix buffer at 4 °C overnight and postfixed for one hour at 4 °C with 1% (v/v) osmium tetroxide before dehydration with increasing concentrations of ethanol (30, 50, 70, 85 and 90% each time for 15 min and 100% twice for 30 min). For TEM observation, samples were embedded with epon (Sigma-Aldrich, MO, US). The ultrathin sections (60 to 80 nm) were stained with uranyl acetate and lead citrate and imaged with a TEM (Tecnai G2 Spirit 120 kV, Thermo FEI). For SEM, samples were dried under vacuum and coated with gold/palladium, then examined by SEM (Nova Nano 450, Thermo FEI).

## Actin polymerization and depolymerization analyses

Actin polymerization/depolymerization analyses were performed according to the guideline of the Actin Polymerization/Depolymerization Assay Kit (ab239724, Abcam). For polymerization assay, prepare background (Buffer G), positive (Buffer P), negative (Latrunculin A, MCE) and test samples (GDYO) on a black 96-well microplate, mixed well with Pyrene-labeled actin monomers and incubated for 15 min. For depolymerization assay, polymerized Actin (F-actin) was made firstly by incubate actin monomers and Buffer P for 1 h. Then positive (Cytochalasin D, MCE), negative (deionized water) and test samples (GDYO) were added respectively. Data acquisition was employed by a multi-well

spectrophotometer (M5, Molecular Device) with kinetic model. To the effect of test sample on actin polymerization or actin depolymerization were represented by calculating ΔRFU. For polymerization assay, $\Delta RFU = (RFU_{Final} - RFU_{Initial})/\Delta t$ and for depolymerization assay, $\Delta RFU = (RFU_{Initial} - RFU_{Final})/\Delta t$. The percentage of polymerization/depolymerization (%) = $\Delta RFU_{Sample}/\Delta RFU_{Positive} \times 100\%$.

## Molecular simulations

**Molecular model and electronic structural calculation.** The electronic structure of bulk and single layered G and GDY were obtained by density functional theory. All calculations were performed with the Vienna ab initio Simulation Package (VASP)[55,56] using projector augmented-wave (PAW) potentials within the PBE exchange-correlation functional[57,58]. The unit cell with one layer periodic slab separated by a 20 Å vacuum region were used to model the single layered carbon materials. The internal atomic positions were fully relaxed until the forces on each atom were less than 0.02 eVÅ−1. An energy cutoff of 500 eV for plane-wave was used. Monkhorst−Pack mesh[59] was used to sample the Brillouin zone: $23 \times 23 \times 15(1)$ for bulk (single layered) G, $6 \times 6 \times 16(1)$ for bulk (single layered GDY). The vdW correction of the GGA functional was included by using the DFT-D2 method in Grimme[60].

The single layered GO and GDYO models were built on the basis of the DFT results and the oxidation groups were introduced. The structures of GO and GDYO models were first optimized at the HF/6-31 G* level and confirmed to be minimal by vibrational analysis using the Gaussian 09 program[61]. The results were used to fit the OPLSS-AA force field parameter set and Partial charges by applying a restrained electrostatic potential (RESP) charge fitting[62]. The models used to calculate molecular orbitals were optimized at B3LYP/6-31 G* level and the molecular orbitals were obtained at the same level.

**Molecular docking.** The single layered GO and GDYO models were built on the basis of the DFT results. The single chain of proteins completed with swiss-model online were used as protein models for the molecular docking simulations. The PDB codes of proteins used for molecular docking simulations as follows: 2OAN (https://doi.org/10.2210/pdb2OAN/pdb), 5AO6 (https://doi.org/10.2210/pdb5AO6/pdb) and 4NEH (https://doi.org/10.2210/pdb4NEH/pdb). All the molecular docking simulations were carried out with Lamarckian Genetic Algorithm (LGA) using Auto-Dock 4.2.6[63]. The grid boxes big enough to cover the entire surface of the peptide were implemented. The docking parameters are as follows: trials of 256 dockings, the population size of 150, the random starting position and conformation, the mutation rate of 0.02, the crossover rate of 0.8, the local search rate of 0.06, and 25 million energy evaluations. The 256 conformations of each docking system with lowest binding energy were selected for the detailed analysis. Final docked conformations were clustered using a tolerance of 2.0 Å root-mean-square deviations (RMSD).

**Molecular dynamics simulations.** All the MD simulations were performed using the Gromacs 5.1 package[64,65] combined with the OPLSS-AA force field[66] and TIP3P solvent model[67]. The NPs are fixed during the simulation and the charge was obtained based on QM calculation. Na+ was used as counter ions to neutralize the systems. After an initial steepest descent minimization of 50000 steps with the stepsize of 0.01 nm, the systems were then heated gradually from 0 to 300 K in the NVT ensemble for 100 ps and 100 ns production MD simulations were performed for the protein and GDYO and GO complexes at a constant temperature of 300 K. The temperature and pressure of the systems were maintained using V-rescale thermostat[68] and isothermal-isobaric ensemble[69], respectively. Particle Mesh Ewald (PME)[70] was employed to deal with the long-range electrostatic interactions under periodic boundary conditions. The coordinates were saved every 10 ps for the subsequent analysis. The contacting

residues are defined as those with their any atom within 5.0 Å of the adsorption surface.

## Reduced representation bisulfite sequencing (RRBS)

$1 \times 10^6$ of OCI-AML3 were pre-treated with GDYO (20 µg/mL) for 24 h and then harvested. Genomic DNA was isolated using the QIAamp DNA blood mini kit (QIAGEN). Libraries were constructed using the Pico Methyl-Seq library prep kit (Zymo Research), and each library was subjected to 150 bp pair-end sequencing on a single lane of a NovaSeq 6000 device (Illumina).

Adaptors and poor quality bases were filtered with Trim Galore (v0.6.0) with default parameters. Clean data were aligned to human reference genome GRCh37 with bismark (v0.21.0)[71] allowing for 1 mismatch. The CpG coverage reports were generated by the bismark_methylation_extractor function. The CpG regions with coverage > 10 in all samples were retained and used for further analysis. The β-value represents the DNA methylation proportion of each CpG, which was calculated by dividing the methylated coverage by total coverage. Principal component analysis (PCA) was performed by R package FactoMineR[72].

## RNA-seq

$1 \times 10^6$ of OCI-AML3 were pre-treated with GDYO (20 µg/mL) for 24 h and then harvested. Total RNA was isolated with Trizol, and then first-strand cDNA synthesis and cDNA libraries were constructed using the NEBNext Ultra™ II RNA Library Prep Kit (New England Biolabs). cDNA quality was determined on Agilent 2100 BioAnalyzer (Agilent Technologies), and then sequenced on NovaSeq 6000 device (Illumina) to obtain pair-end 150 bp reads.

Reads were trimmed by trim galore (0.4.0) with default parameters to remove the reads containing adapters and showing low quality. Raw data were trimmed by Trimmomatic (v0.38)[73]. Clean data were aligned to the human reference genome GRCh37 by HISAT2 (v2.1.0) with default setting[74]. HTSeq (v0.11.2)[75] was used to calculate genes counts with parameter "--mode intersection-strict --stranded no --minaqual 1". Differential expressed genes (DEGs) were defined with a fold change >2 and false discovery rate (FDR) < 0.05 by edgeR (v3.22.3)[76]. Database for Annotation, Visualization and Integrated Discovery (DAVID) v6.8 web tool (https://david.ncifcrf.gov/) was used to perform GO and KEGG enrichment analyses with a significance of $P < 0.05$[77]. Gene set enrichment analysis (GSEA) was performed to explore whether identified sets of genes showed significant differences between two groups using the C2-KEGG collection from MSigDB[78].

## Analysis of published data

The raw counts data of RNA-seq of AML cell lines or BM cells derived from AML patients were download from Cancer Cell Line Encyclopedia (CCLE, https://depmap.org/portal/download/)[14] and The Cancer Genome Atlas (TCGA, https://www.cancer.gov/tcga) database respectively. As described in RNA-seq analysis section, DEGs between *DNMT3A* with or without mutation were determined using edgeR. DNA Methylation Array data (HumanMethylation450K), which derived from same cohorts as above-mentioned RNA-seq data, were download from TCGA. DNA methylation coverage was transformed to β-value. Then the β-value and normalized counts from RNA-seq were used to calculate DNA methylation and gene expression correlation by Pearson test.

RNA-seq data of LSKs from the *Dnmt3a*^R882H mice were downloaded from previous study[15]. RNA-seq data of HSCs from the *Dnmt3a* knockout mice were downloaded from previous study[5] with GEO accession number GSE50793. RNA-seq data of HSCs, AML blasts and LSCs from the patient samples were downloaded from previous study[44] with GEO accession number GSE74912. RNA-seq data of human AML cells and HSCs were downloaded from previous study[79] with GEO accession number GSE13159 and GSE42519. RNA-seq data of 17 mouse hematopoietic cells were downloaded from previous study[80] with GEO accession number GSE87556.

## Statistical analysis

Each experiment was repeated independently at least three times. All the data are represented as mean ± standard deviation (SD). Student's *t*-test was used for statistical analysis of two groups, and one-way ANOVA followed by Tukey's post-hoc tests was used for comparisons among multiple groups differences among multiple groups. Survival data were determined for every group by the Kaplan–Meier method and compared by the log-rank (Mantel–Cox) test. The *P* value was calculated using GraphPad Prism 8, those <0.05 were considered significant.

## Reporting summary

Further information on research design is available in the Nature Research Reporting Summary linked to this article.

## Data availability

Source data are provided with this paper. Source data is available for all main figures and supplementary figures 1, 6, 7, 9, 11–14, 20, 22, 24, 29, 34, 35, 37–40 in the associated source data file. All the sequencing data are accessible at GEO database with accession number GSE190874. Survival analysis of AML patients with high and low gene sets expression were isolated from GEPIA2 (http://gepia2.cancer-pku.cn/#survival). Expressions of ITGB2 and MRC2 in human normal hematopoietic cells were isolated from The Human Protein Atlas: ITGB2 (https://www.proteinatlas.org/ENSG00000160255-ITGB2/immune+cell) and MRC2 (https://www.proteinatlas.org/ENSG00000011028-MRC2/immune+cell). Source data are provided with this paper.

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

## Acknowledgements

This work was supported by grants from the National Key R&D Program of China, Stem Cell and Translation Research (2018YFA0109300), the National Natural Science Foundation of China (NSFC) (82222003, 82000149, 81870080, 91949115 and 31971318), Zhejiang Province Science Foundation for Distinguished Young Scholars (LR19H080001) and Exploration Project (LQ21H180006), the China Postdoctoral Science Foundation (CPSF) (2021M702853), the Leading Innovative and Entrepreneur Team Introduction Program of Zhejiang (2020R01006), Major instrument project of National Natural Science Foundation of China (22027810), Strategic Priority Research Program of Chinese Academy of Sciences (XDB36000000), the Innovative Research Groups of the NSFC (11621505), the Key-Area Research and Development Program of Guangdong Province (2019B090917011, 2020B0101020001), and the NSFC (21790050, 21790053, 22071251 and 21875258) to Huibiao Liu. We thank Dr. Hangjun Wu and Dr. Jiansheng Guo in the Center of Cryo-Electron Microscopy (CCEM), Zhejiang University for their technical assistance on transmission and scanning electron microscopy. We thank Ms. Wei Yin from the Core Facilities, Zhejiang University School of Medicine for her technical support on confocal laser scanning microscopy.

## Author contributions

Q.W. designed and performed all the experiments, Q.W. and Y. Liu analyzed data and wrote the manuscript. H.W., K.H. and X.S. performed quantum mechanical calculations and molecular dynamics simulations. P.J., W.Q. and H. Lu performed bioinformatics analysis of RNA-seq/RRBS and published data. Y.H. and X.Z. constructed AML mouse model. J.L., L.J. M.Z., M.T. and G.B. provided technical assistance. S.L. and X.W. assisted to data analysis. M.Y. and L.Y. performed characterizations of GDYO and GO nanosheets. Z.X., H. Liu, Y. Li, and Y.Z. provided the GDYO and GO nanosheets. P.Q. and C.C. supervised the overall project and co-wrote the manuscript. All authors contributed to reading and editing the manuscript.

## Competing interests

The authors declare no competing interests.

## Additional information

[1]Center of Stem Cell and Regenerative Medicine, and Bone Marrow Transplantation Center of the First Affiliated Hospital, Zhejiang University School of Medicine, Hangzhou 310058, China. [2]Liangzhu Laboratory, Zhejiang University Medical Center, 1369 West Wenyi Road, Hangzhou 311121, China. [3]Institute of Hematology, Zhejiang University & Zhejiang Engineering Laboratory for Stem Cell and Immunotherapy, Hangzhou 310058, China. [4]CAS Key Laboratory for Biomedical Effects of Nanomaterials and Nanosafety and CAS Center for Excellence in Nanoscience, National Center for Nanoscience and Technology of China, Beijing 100190, China. [5]University of Chinese Academy of Sciences, Beijing 100049, China. [6]The GBA National Institute for Nanotechnology Innovation, Guangzhou 510700, China. [7]Laboratory of Theoretical and Computational Nanoscience, CAS Center for Excellence in Nanoscience, National Center for Nanoscience and Technology, Beijing 100190, China. [8]Institute of Brain and Cognition, Zhejiang University City College School of Medicine, Hangzhou 310015, China. [9]The MOE Frontier Research Center of Brain & Brain-Machine Integration, Zhejiang University School of Brain Science and Brain Medicine, Hangzhou 310058, China. [10]CAS Key Laboratory for Biomedical Effects of Nanomaterials and Nanosafety, Institute of High Energy Physics and National Center for Nanoscience and Technology of China, Chinese Academy of Sciences, Beijing 100049, China. [11]Beijing National Laboratory for Molecular Sciences, Key Laboratory of Organic Solids, Institute of Chemistry, Chinese Academy of Sciences, Beijing 100190, China. [12]These authors contributed equally: Qiwei Wang, Ying Liu, Hui Wang, Penglei Jiang, Wenchang Qian. ✉e-mail: chenchy@nanoctr.cn; axu@zju.edu.cn

