## [Peer Review File · Nature Communications]

Graphdiyne oxide nanosheets display selective anti-leukemia efficacy against DNMT3A-mutant AML cellsREVIEWER COMMENTS

Reviewer #1 (Remarks to the Author):

This is an interesting and informative paper on defining the critical role of graphdiyne oxide nanosheets in disrupting DNMT3A-mutant acute myeloid leukemia cell functions through various mechanism including their direct interacts with important receptors (i.e., integrin $\beta 2$ and c-type mannose) that are highly expressed in these cells. The authors could successfully correlate the observed in vitro efficacy of graphdiyne oxide nanosheets with their in vivo therapeutic role in mouse model of acute myeloid leukemia.

The authors are encouraged to improve the robustness of the presented data and interpretation of the results, based on the following comments:

1- The authors used 20% FBS in the cell culture medias for the in vitro cell studies. This means that the surface of the incubated graphdiyne oxide nanosheets had been covered by the FBS proteins through the protein corona formation mechanism. Therefore, the observed therapeutic efficacy of the graphdiyne oxide nanosheets (e.g., their interaction with cell receptors and actin cytoskeletal structure) may also have important relation to the type and decoration of the participated proteins in the corona shell at the surface of the nanosheets. Conducting the two complementary experiments would shed more light on the actual mechanism of actions of the nanosheets: i) conducting the nanosheet-cell interactions in serum free media; and ii) incubation of nanosheet with either conditioned cell media or 20% FBS solution and analyze the protein corona profiles. The proteomics analysis on the cells in these conditions (in comparison with the presented data in Figure 4) can make the authors' claims more robust.

2- For correlating in vitro and in vivo results in a meaningful and robust way, the authors need to use mice plasmas to make corona coated nanosheets for in vitro studies. In addition, to better understand the in vivo mechanism of actions of nanosheets, the protein corona profiles of nanosheets (after interaction with mouse plasmas) should be fully analyzed.

3- The authors need to conduct some toxicological analysis at the cellular level to show that the actin disrupting role of graphdiyne oxide nanosheets is not applicable in other sensitive cells (e.g., neurons).

4- The presented molecular dynamics (MD) based data is interesting, but lacked the role of nano-bio interfaces (protein corona) on the surface of the nanosheets. The obtained information on the protein corona of nanosheets (please see the first comment for details) can provide the require info for the MD experts to see whether there is specific interactions wit the receptors and protein-sections (selected corona proteins with critical potential to interact with the receptors).

Reviewer #2 (Remarks to the Author):

In this manuscript, Wang et al. showed that cell adhesion-related genes are enriched specifically in AML cells with DNMT3A mutations and that the carbon nanomaterial graphdiyne oxide (GDYO) displayed an anti-leukemia effect specifically against DNMT3A mutant AML cells. The findings are potentially significant with therapeutic implications.

I have the following suggestions:

1. While not an expert on nanomaterials, I find the proposed mechanisms by which GDYO exerts its effect confusing. On the one hand, the authors showed that GDYO interacts with integrin $\beta 2$ and MRC2, which are highly expressed in DNMT3A-mutant AML cells, and binds actin and prevents actin polymerization. On the other, the authors showed that GDYO treatment, albeit not affecting overall DNA methylation, alters DNA methylation at multiple loci, especially genes involved in focal adhesion

and actin cytoskeleton. At which level (protein or DNA methylation) does GDYO act? It makes sense to me that GDYO binds cell adhesion molecules and actin. Is the methylation change at genes involved in focal adhesion and actin cytoskeleton a secondary effect, thus forming some sort a feed-forward loop? The authors ought to provide experimental evidence for the significance of DNA methylation change in gene expression or downplay that as a possible mechanism of GDYO action.

2. Fig 1a, how many common cell adhesion-related genes were identified from both TCGA and CCLE databases? Do they include integrin β 2 and MRC2? Validation of their upregulation experimentally (e.g. RT-qPCR, IF) would strengthen the conclusion that cell adhesion molecules are enriched in DNMT3A-mutant AML cells.

3. Minor issues: a) Page 8. The sentence "Overall, these data suggest that GDYO these data suggest that GDYO is more..." needs to be corrected by deleting "these data suggest GDYO". b) Page 13. In the sentence "...when treated with daunorubicin, like due to...", like should be "likely".

Responses to the Reviewers' Comments

We greatly appreciate the time and effort of reviewers in providing constructive critiques of our initial manuscript. We are grateful to all the reviewers for their positive comments regarding the rigor and clarity of our manuscript. All reviewers value our study as comprehensive and thorough with compelling experimental evidence, and have highlighted the strength of our work and the contribution to GDYO-based therapeutics for DNMT3A-mutant AML. Reviewer #1 remarked, "This is an interesting and informative paper on defining the critical role of graphdiyne oxide nanosheets in disrupting DNMT3A-mutant acute myeloid leukemia cell functions". Reviewer #2 pointed out "The findings are potentially significant with therapeutic implications".

Their valuable comments and suggested experiments help very much to improve the quality of our manuscript. Based on their constructive comments, we have made extensive revisions of the manuscript with additional experimental studies, including providing proteomics analysis of GDYO coated with FBS and mouse plasma proteins, detecting their respective *in vitro* anti-leukemia effects, conducting toxicological analysis of GDYO in hematopoietic stem cells and neurons and analyzing the expression level of cell adhesion molecules in DNMT3A-mutant AML cells. These additional data greatly support our model that GDYO turns on its anti-leukemia efficacy by interacting with ITGB2 and MRC2 expressed on membrane of DNMT3A-mutant AML cells. Below we present a more detailed accounting of the revisions outlined as a point-by-point response to each comment by each reviewer.

Reviewers' comments:

Reviewer #1: *This is an interesting and informative paper on defining the critical role of graphdiyne oxide nanosheets in disrupting DNMT3A-mutant acute myeloid leukemia cell functions through various mechanism including their direct interacts with important receptors (i.e., integrin β 2 and c-type mannose) that are highly expressed in these cells. The authors could successfully correlate the observed in vitro efficacy of graphdiyne oxide nanosheets with their in vivo therapeutic role in mouse model of acute myeloid leukemia.*

Answer: We thank the reviewer for his/her positive general remarks on our study. According to the reviewer's suggestions, we have made the corresponding changes in the revised manuscript.

The authors are encouraged to improve the robustness of the presented data and interpretation of the results, based on the following comments:

1- *The authors used 20% FBS in the cell culture medias for the in vitro cell studies. This means that the surface of the incubated graphdiyne oxide nanosheets had been covered by the FBS proteins through the protein corona formation mechanism. Therefore, the observed therapeutic efficacy of the graphdiyne oxide nanosheets (e.g., their interaction with cell receptors and actin cytoskeletal structure) may also have important relation to the type and decoration of the participated proteins in the corona shell at the surface of the nanosheets. Conducting the two complementary experiments would shed more light on the actual mechanism of actions of the nanosheets: i) conducting the nanosheet-cell interactions in serum free media; and ii) incubation of nanosheet with either conditioned cell media or 20% FBS solution and analyze the protein corona profiles. The proteomics analysis on the cells in these*

conditions (in comparison with the presented data in Figure 4) can make the authors' claims more robust.

Answer: The reviewer raised a very important question. As the first of the two complementary experiments, we had indeed screened for binders of GDYO in serum-free media (Line 590-592 of Methods part in the primary manuscript), and identified ITGB2 and MRC2 as the GDYO-binding proteins in serum-free media. Secondly, we agreed with the reviewer that the surface of the incubated GDYO nanosheets had been covered by the serum proteins through the protein corona formation mechanism. Hence, we have now incubated GDYO nanosheets with 20% FBS solution, and analyzed the protein profiles by performing the LC-MS/MS (GDYO^B, Fig. 4k-m and Supplementary Table 2). Interestingly, we found that the absorbed proteins in FBS were mainly extracellular matrix proteins, such as collagen and fibronectin (Fig. 4n and Fig. S23). We have now added these data in the revised manuscript, which makes our conclusion more robust.

Figure 4k. Sample preparation diagram of to profile proteins absorbed on GDYO^B/GDYO^M and to detect the in vitro cytotoxicity towards OCI-AML3 in serum-free media. **Figure 4l.** Protein content quantification of proteins absorbed on GDYO^B/GDYO^M. **Figure 4m.** Venn diagram of protein species of proteins absorbed on GDYO^B/GDYO^M. **Figure 4n.** Top 15 abundant proteins absorbed on GDYO^B/GDYO^M.

2- For correlating in vitro and in vivo results in a meaningful and robust way, the authors need to use mice plasmas to make corona coated nanosheets for in vitro studies. In addition, to better understand the in vivo mechanism of actions of nanosheets, the protein corona profiles of nanosheets (after interaction with mouse plasmas) should be fully analyzed.

Answer: We appreciate the critical and insightful comments from the reviewer. According to the reviewer's suggestions, we have also analyzed the protein profiles through incubation of GDYO nanosheets with mouse plasma solution (GDYO^M, Fig. 4k-n, Fig. S23 and Supplementary Table 2). We found that the quantity of total protein of GDYO^M was significantly larger than that of GDYO^B (Fig. 4l), while GDYO^B interacted with more proteins (Fig. 4m). This might be due to the fact that fibrinogen is present in plasma whereas not in serum, which account for about 37% of the protein abundance in GDYO^M, including fibrinogen alpha chain (Fga), fibrinogen beta chain (Fgb) and fibrinogen gamma chain (Fgg) (Fig. 4n). For correlating the in vitro and in vivo results, we compared the anti-leukemia efficacies against OCI-AML3 of bare GDYO, GDYO^B and GDYO^M in serum-free media (Fig. 4o-r, Fig. S24), and found that GDYO^M exhibited more decreased cell viability, increased apoptosis and cell differentiation, and showed more reduced number of LSCs and colony formation unit (CFU) than bare GDYO in OCI-AML3, whereas GDYO^B did not. The above results might partially explain the in vivo therapeutic efficacy of GDYO.

Figure 4o-p. Cell viability (o) and apoptotic rates (p) in OCI-AML3 treated with 20 µg/mL GDYO/GDYO^B/GDYO^M for 48 h. **Figure 4q-r.** Frequency (q) and representative flow images (r) of leukemic progenitor cells and leukemic stem cells in OCI-AML3 treated with 20 µg/mL GDYO/GDYO^B/GDYO^M for 72 h.

3- The authors need to conduct some toxicological analysis at the cellular level to show that the actin disrupting role of graphdiyne nanosheets is not applicable in other sensitive cells (e.g., neurons).

Answer: We thank the reviewer for pointing this out. According to the reviewer's suggestions, hematopoietic stem cells (HSCs) are the most sensitive cells in the blood system. Thus, we have conducted 28 days long-term co-culture of GDYO nanosheets and mouse HSCs, and found that long-term GDYO co-culture neither decreased cell number nor induced apoptosis of HSCs (Fig. S40d-f). Moreover, as the reviewer suggested, we also used other types of sensitive cells (e.g., neurons), and revealed that high concentration of GDYO also did not induce apoptosis in neuron cells (Fig. S40g-h). Together, these data would strengthen the biosafety profile of GDYO.

Figure S40d-h. Biosafety assessment of GDYO. **d**, Representative flow cytometry images of LSK and LT-HSC after GDYO co-incubation for 28 d. **e, f**, Absolute number (**e**) and apoptotic frequency (**f**) of LT-HSC after GDYO co-incubation for 28 d. **g, h**, Representative flow cytometry images (**g**) and apoptotic frequency (**h**) of human neuron cell line SH-S5Y5 after GDYO co-incubation for 48 and 72 h.

4- *The presented molecular dynamics (MD) based data is interesting, but lacked the role of nano-bio interfaces (protein corona) on the surface of the nanosheets. The obtained information on the protein corona of nanosheets (please see the first comment for details) can provide the require info for the MD experts to see whether there is specific interactions with the receptors and protein-sections (selected corona proteins with critical potential to interact with the receptors).*

Answer: The reviewer raises an interesting question. Previous studies have reported that fibrinogen is one of ligands of the ITGB2-containing integrins including integrin $\alpha_x\beta_2$ and $\alpha_m\beta_2$, integrin $\alpha_x\beta_2$ recognizes the sequence G-P-R in fibrinogen alpha-chain, and integrin $\alpha_m\beta_2$ recognizes P1 and P2 peptides of fibrinogen gamma chain (Wright SD, *et al*, *PNAS*, 1988, 85: 7734-38; Loike JD, *et al*, *PNAS*, 88:1044-48). Moreover, MRC2 is a receptor of collagen (Engelholm LH, *et al*, *J Cell Biol* 2003, 160: 1009-15). Since both the fibrinogen and collagen are very large-size extracellular proteins and lack relevant structural informations, it's very hard and laborious to perform the MD analysis. Instead, to address this, we co-incubated GDYO^B/GDYO^M with plasma membrane protein solution of OCI-AML3 and performed the pull-down experiments (Fig. 4i). The pull-down results showed that GDYO^B had increased binding affinity to MRC2 but no change to ITGB2 relative to bare GDYO, while GDYO^M had significantly increased binding affinity to both receptors (Fig. 4j). These results provide explanations for the different *in vitro* anti-leukemia efficacies of GDYO^B and GDYO^M in OCI-AML3, and indicate that the binding of GDYO to both ITGB2 and MRC2 is indispensable for its anti-leukemia effect.

Figure 4i. Sample preparation diagram of to prepare GDYO^B/GDYO^M and to detect the interacted receptors. **Figure 4j.** Antibody-guided immunoblot for binding ability between ITGB2/MRC2 and GDYO/GDYO^B/GDYO^M.

Reviewer #2: *In this manuscript, Wang et al. showed that cell adhesion-related genes are enriched specifically in AML cells with DNMT3A mutations and that the carbon nanomaterial graphdiyne oxide (GDYO) displayed an anti-leukemia effect specifically against DNMT3A mutant AML cells. The findings are potentially significant with therapeutic implications.*

Answer: We thank the reviewer for pointing out the quality and importance of our study. Now we provide further pertinent experimental evidences to address these concerns, which greatly help improve the quality and strengthen the conclusions of our study.

I have the following suggestions:

- 1. While not an expert on nanomaterials, I find the proposed mechanisms by which GDYO exerts its effect confusing. On the one hand, the authors showed that GDYO interacts with integrin $\beta 2$ and MRC2, which are highly expressed in DNMT3A-mutant AML cells, and binds actin and prevents actin polymerization. On the other, the authors showed that GDYO treatment, albeit not affecting overall DNA methylation, alters DNA methylation at multiple loci, especially genes involved in focal adhesion and actin cytoskeleton. At which level (protein or DNA methylation) does GDYO act? It makes sense to me that GDYO binds cell adhesion molecules and actin. Is the methylation change at genes involved in focal adhesion and actin cytoskeleton a secondary effect, thus forming some sort a feed-forward loop? The authors ought to provide experimental evidence for the significance of DNA methylation change in gene expression or downplay that as a possible mechanism of GDYO action.*

Answer: We appreciate this critical and insightful comments from the reviewer. Our bisulfite sequencing in OCI-AML3 cells suggested that GDYO did not directly influence DNMT3A methyltransferase activity (Fig. S25a-e). We agree with the reviewer that the methylation change at genes involved in focal adhesion and actin cytoskeleton is a secondary effect, and we did lack evidences about why cells respond at the level of DNA methylation. Therefore, to downplay that DNA methylation alterations at multiple loci as a possible mechanism of GDYO action, we have now removed relevant data for differentially methylated sites in the revised manuscript, which would make our conclusion more clear to readers.

- 2. Fig 1a, how many common cell adhesion-related genes were identified from both TCGA and CCLE databases? Do they include integrin $\beta 2$ and MRC2? Validation of their upregulation experimentally (e.g. RT-qPCR, IF) would strengthen the conclusion that cell adhesion molecules are enriched in DNMT3A-mutant AML cells.*

Answer: We appreciate the reviewer's suggestion. We listed the up-regulated cell adhesion-related genes from both TCGA and CCLE databases and found three overlapped genes including ITGB2 (**Fig. S1a**). MRC2 was not found, because MRC2 is not a cell adhesion molecule and normally acts as an endocytic receptor (**Sheikh H, et al, J Cell Sci 2000, 113(6): 1021-32**). Moreover, we have validated the expression of three overlapped genes by RT-qPCR, and the results showed that all three genes were up-regulated in human DNMT3A-mutant AML cell lines (**Fig. S1b**).

Figure S1. Up-regulated genes related to “Cell Adhesion” in *DNMT3A*-mutant AML. a, Venn diagram of up-regulated genes in *DNMT3A*-mutant AML cells isolated from both AML patients (TCGA: *DNMT3A^{mut}*, $n = 36$; *DNMT3A^{WT}*, $n = 115$) and AML cell lines (CCLE: *DNMT3A^{mut}*, $n = 4$; *DNMT3A^{WT}*, $n = 30$). **b**, Q-PCR verification of three overlapped genes of TCGA and CCLE databases in human AML cell lines.

3. Minor issues:

a) Page 8. The sentence “Overall, these data suggest that GDYO these data suggest that GDYO is more...” needs to be corrected by deleting “these data suggest GDYO”.

Answer: We appreciate the reviewer’s comment and now delete the repeated sentence in the revised manuscript.

b) Page 13. In the sentence “...when treated with daunorubicin, like due to...”, like should be “likely”.

Answer: We appreciate the reviewer's comment and now correct them in the revised manuscript.

REVIEWERS' COMMENTS

Reviewer #1 (Remarks to the Author):

The authors have conducted the requested additional experiments; with the new data, the outcomes of the revised manuscript is now robust; and the in vitro and in vivo results are well correlated. I'd publish this interesting manuscript.

Reviewer #2 (Remarks to the Author):

The issues raised by me have been satisfactorily addressed in the revised manuscript.